# H2MV (v1.0): Global Physically-Constrained Deep Learning Water Cycle Model with Vegetation

Zavud Baghirov[1,2], Martin Jung[1], Markus Reichstein[1,5], Marco Körner[2,3], and Basil Kraft[1,4]

[1]Department of Biogeochemical Integration, Max Planck Institute for Biogeochemistry, Jena, Germany
[2]Department of Aerospace and Geodesy, TUM School of Engineering and Design, Technical University of Munich (TUM), Germany
[3]Munich Data Science Institute, Technical University of Munich (TUM), Munich, Germany
[4]ETH Zurich, Environmental Systems Science, Zurich, Switzerland
[5]ELLIS Unit Jena at Michael-Stifel-Center Jena for Data-driven and Simulation Science, Jena, Germany

**Correspondence:** Zavud Baghirov (zbaghirov@bgc-jena.mpg.de)

**Abstract.** The proposed hybrid hydrological model with vegetation (H2MV) uses dynamic meteorology and static features as input to a long short-term memory (LSTM) to model uncertain parameters of process formulations that govern water fluxes and states. In the hydrological model, vegetation states are represented by the fraction of absorbed photosynthetically active radiation (fAPAR), and soil storage capacity ($SM_{\max}$), which depends on effective rooting depth besides soil properties. $SM_{\max}$ and fAPAR are both learned and predicted by the neural networks directly. These parameters have an explicit role to model soil moisture (SM) storage and the partitioning of evapotranspiration (ET). The model is optimised concurrently against global observations and observation-based data of terrestrial water storage (TWS) anomalies, fAPAR, snow water equivalent (SWE), ET and gridded runoff in a 10-fold cross-validation setup. To this end, we infer where the model is under-constrained such that different processes could explain the observational constraints in the model due to equifinality. The model reproduces the observed patterns of global hydrological components and fAPAR, while emergent patterns of runoff ratio, evaporative fraction, and T/ET are consistent with our current understanding. Despite robustly predicted temporal patterns of TWS anomalies, we found that the mean soil moisture state is not well constrained causing uncertainty of mean TWS. This emphasizes the importance of $SM_{\max}$ and the necessity for associated enhanced constraints. The proposed model is open-source, and has a highly flexible and modular structure to facilitate future integration of carbon and energy cycles, advancing toward a hybrid land surface model.

## 1 Introduction

Global hydrological models (GHMs) play a foundational role to understand Earth's water resources on a large scale. They provide important insights into predicting extreme events, managing water scarcity, and planning sustainable water resources under changing climate (Zhang et al., 2023).

GHMs simulate key hydrological processes including evapotranspiration, runoff, and soil moisture. They employ process-based models (PBMs), which are abstracted representations of the processes controlling water movement and distribution

within a hydrological system. PBMs rely on established physical principles such as the conservation of mass and energy (Fatichi et al., 2016). By adhering to these fundamental laws of physics, PBMs offer hydrologists a unique approach to studying the global hydrological system.

25    Despite their utility, PBMs encounter significant challenges. Some of the process knowledge can be incomplete and the theories and assumptions underpinning model development can sometimes be subjective, leading to uncertainties in parameter estimations within GHMs (Nearing et al., 2021). Additionally, PBMs were typically not designed to fully harness the growing Earth observation (EO) data, which can limit their capacity to capture unknown or unexpected processes (Shen et al., 2018).

Machine learning (ML), particularly deep learning (DL), effectively addresses the challenge of learning from and utilizing 30    large amounts of observational data. DL can significantly decrease the requirement for domain expertise, operate with much fewer assumptions, and possess the capacity to unveil unexpected processes due to their versatile internal architectures (LeCun et al., 2015). DL models have been garnering increased attention in hydrology and have repeatedly been shown to outperform physics-based models (Nearing et al., 2021; Sit et al., 2020). However, DL models come with noteworthy disadvantages. In contrast to PBMs, DL models offer no assurance of respecting the laws of physics, even when delivering outstanding 35    predictions. Therefore, interpreting the learned internal functions of deep learning models becomes highly challenging (Alain and Bengio, 2016; Shwartz-Ziv and Tishby, 2017) with potentially implausible responses learned such that trust in models when applied on new data is limited (Geirhos et al., 2020).

Hybrid modeling aims to address this challenge. This approach facilitates the design of models that preserve certain process representations of a PBM, while incorporating the ability to learn uncertain components through DL from observations 40    (Reichstein et al., 2019; Shen et al., 2023).

Recent studies have been exploring the integration of process knowledge into machine learning models to better constrain uncertain processes with hybrid approaches. For instance, Zhao et al. (2019) developed a hybrid model that merges a NN with an evapotranspiration model to estimate latent heat flux, ensuring it adheres to the conservation of energy principle. This model performed better in extrapolating beyond the data range of the training set, compared to a more data-driven model. Similarly, 45    ElGhawi et al. (2023) combined NN with a mechanistic latent heat flux model to estimate the surface and aerodynamic resistances of vegetation. While their model successfully estimated latent heat flux, it faced the challenge of equifinality. To address this, they applied both theoretical and data constraints. In a comparable effort, Koppa et al. (2022) utilized a process-based model of terrestrial evaporation alongside NN to estimate transpiration stress. Zhong et al. (2023) integrated deep learning with a hydrological model to estimate runoff changes, demonstrating that this approach enhances the reliability of projections 50    in permafrost-affected mountain headwaters. Similarly, Bennett and Nijssen (2021) combined neural networks with a process-based hydrological model to simulate turbulent heat fluxes, concluding that this method offers advantages over both purely process-based models and purely machine learning-based estimations. Additionally, Bhasme et al. (2022) merged neural networks with a conceptual hydrological model to effectively estimate evapotranspiration and streamflow in regional catchments.

Studies by Kraft et al. (2020) and Kraft et al. (2022) employed the hybrid method in global hydrological modeling. They 55    utilized a dynamic neural network (NN), specifically a Long Short-Term Memory (LSTM) model (Hochreiter and Schmidhuber, 1997), to estimate coefficients of a simple conceptual hydrological PBM. The hybrid model is trained end-to-end, i.e. the

feedback from the PBM is used to optimize the weights of the NNs, and simulates the dynamics of evapotranspiration, runoff, and water storages. The study employed observational products of TWS variations, snow, ET, and runoff to constrain (i.e., to calibrate) the model. However, the model has certain limitations. For instance, soil moisture was represented implicitly by a cu-
mulative water deficit term, evapotranspiration components, transpiration, soil, and interception evaporation were not resolved, and the role of vegetation, an important aspect in global hydrological modeling (Trautmann et al., 2022), was not explicitly accounted for.

We present here the global hybrid hydrological model with vegetation (H2MV) that explicitly represents two pivotal properties of vegetation: The maximum soil water storage capacity $SM_{\mathrm{max}}$ and fraction of absorbed photosynthetically active
radiation (fAPAR), extending previous work by Kraft et al. (2022). The $SM_{\mathrm{max}}$ is a crucial parameter that governs water availability for plants and thus the interactions between water and carbon cycles. While Kraft et al. (2022) estimated the cumulative soil water deficit as a proxy for soil moisture and without any physical limit to the maximum deficit, the implementation of $SM_{\mathrm{max}}$ adds a relevant conceptual constraint and facilitates an explicit representation of plant available soil moisture. This parameter is currently not observable on a global scale and the spatial patterns of $SM_{\mathrm{max}}$ remain highly uncertain (Stocker
et al., 2023). Vegetation state is represented by directly estimating the daily patterns of fAPAR, constrained against satellite observations. The inclusion of fAPAR in the model is relevant for modelling ET components (transpiration, soil and interception evaporation).

In this study, we also address the prevalent issue of equifinality, which is one of the main limitations in PBM in general (Beven and Freer, 2001; Beven, 2006), and hybrid modeling in particular (Kraft et al., 2022). Equifinality is the condition where
different combinations of model parameters or different model configurations yield similar results, making it challenging to identify a single 'correct' model. This problem is exacerbated in the context of hybrid models that incorporate NNs due to their inherent flexibility. The structure of these models imposes fewer constraints, potentially complicating the equifinality issue further. Concurrently, traditional methods for assessing parameter correlations and equifinality fall short when applied to hybrid models. This inadequacy stems from the unique complexities and characteristics of hybrid models, necessitating the
exploration of alternative approaches for the effective assessment of the equifinality problem. Therefore, we develop a simple approach for the quantification of parameter robustness, which allows diagnosing model shortcomings. The equifinality of estimated parameters is assessed using a 10-fold cross-validation (CV) approach. Ten different models are trained with varied training and validation sets, and a simple metric is used to quantify equifinality in the estimated parameters.

For transparency and reproducibility, the model is designed in a modular structure and shared with the community. Compre-
hensive documentation accompanies the code, which is openly shared on a public repository. This commitment to transparency encourages open-source collaboration and ensures full reproducibility for specifically developing the model further towards a global hybrid land-surface model.

Specifically, this work has the following key objectives:

   – Extend previous work by 1) explicitly representing vegetation, constrained by satellite observations, 2) partitioning ET
into transpiration, soil evaporation, and interception evaporation, and 3) improve representations of soil moisture by an improved parameterization via maximum soil moisture $SM_{\mathrm{max}}$.

    – Identify equifinality by quantifying parameter robustness.

    – Ensure transparency and model reproducibility.

## 2 Methods and datasets

### 2.1 Datasets

We use meteorological time series data—specifically precipitation, net radiation, and air temperature—as temporal inputs (forcing) for our model. In addition to these temporal inputs, we incorporate static features such as soil properties, land cover fractions, elevation, and wetlands as static inputs.

Our model is optimized against observations of terrestrial water storage, fAPAR, and snow water equivalent, as well as observation-based estimations of evapotranspiration and runoff. We refer to these observations as "constraints", as they confine the model's behavior to the observed patterns.

Table 1 shows the detailed information about the used datasets. All meteorological forcing and model constraints were aggregated to 1°spatial resolution. The spatial resolutions of static inputs were aggregated to 1/30°. We use compressed representations of the original static input that was preprocessed in a separate modeling framework (for details Kraft et al. (2022) can be referred). Meteorological forcing and SWE are kept in the native daily temporal resolutions, while monthly temporal resolution is used for the rest of the model constraints.

Temporal coverage of the data we use vary:

    – Meteorological forcing (all): 2001 to 2019 (19 years of daily data)

    – TWS: 2001 to 2017 (17 years of monthly data)

    – fAPAR: 2001 to 2019 (19 years of monthly data)

    – SWE: 2001 to 2018 (18 years of daily data)

    – ET: 2001 to 2015 (15 years of monthly data)

    – Runoff: 2001 to 2019 (19 years of monthly data)

### 2.2 H2MV

This section outlines the workflow of our hybrid model, which integrates modeled hydrological processes with NN within an end-to-end framework, as illustrated in Fig. 2. The model is composed of two main parts: a dynamic sub-module and a static sub-module.

In the dynamic sub-module, we use an LSTM model to process both dynamic meteorological data and static features. The LSTM model is designed to learn temporal parameters (coefficients) that are physically interpretable, aiding in the prediction of parameters that are typically uncertain due to the lack of direct observations or incomplete process knowledge. These

**Table 1.** Datasets used: meteorological forcing, static inputs and model constraints. The resolution column shows the original resolutions.

| Name | Resolution | | Data | Reference |
| | Spatial | Temporal | | |
| --- | --- | --- | --- | --- |
| **Meteorological forcing** | | | | |
| Precipitation | $1°$ | Daily | GPCP 1dd v1.2 | Huffman et al. (2016) |
| Net radiation | $1°$ | Daily | CERES SYN1deg Ed4A | Wielicki et al. (1996), Doelling (2017) |
| Air temperature | $0.5°$ | Daily | CRUNCEP v8 | Harris et al. (2014), Viovy (2018) |
| **Static data** | | | | |
| Soil properties | $1/120°$ | - | Soil grids v2 | Hengl et al. (2017) |
| Land cover fractions | $1/360°$ | - | Globland30 v1 | Chen et al. (2015) |
| Digital elevation model | $1/120°$ | - | GTOPO | Center (1997) |
| Wetlands | $1/240°$ | - | Tootchi | Tootchi et al. (2019) |
| **Model constraints** | | | | |
| Terrestrial water storage | $0.5°$ | Monthly | GRACE Tellus JPL RL06M v1 | Watkins et al. (2015) |
| fAPAR | $500m$ | 8 daily | MOD15A2H | Myneni et al. (2015) |
| Snow water equivalent | $0.25°$ | Daily | GlobSnow v2 | Takala et al. (2011), Luojus et al. (2014) |
| Evapotranspiration | $0.5°$ | Monthly | FLUXCOM v1 | Tramontana et al. (2016), Jung et al. (2019) |
| Runoff | $0.5°$ | Monthly | GRUN v1 | Ghiggi et al. (2019) |

predictions are then utilized within a conceptual hydrological model to estimate water fluxes and storages, with some estimates being constrained by available observational data.

The static sub-module processes static features through a fully connected NN to determine spatially varying parameters. This approach allows for the estimation of parameters that do not change over time but vary across different spatial locations.

Together, these sub-modules enable H2MV to provide a comprehensive understanding of hydrological processes by leveraging both dynamic and static data sources.

### 2.2.1 Hydrological model

In this section, we present the conceptual model of the hydrological cycle, offering a high-level overview of the modeled processes as depicted in Fig. 1. We focus on describing the key hydrological processes and the underlying logic that changed compared to Kraft et al. (2022). For a comprehensive understanding, the full model is detailed in Appendix A.


In the equations below, parameters denoted with the superscript $<s, t>$ show variables varying both in space ($s$) and time ($t$), while those marked with the superscript $<s>$ refer solely to spatial variation. Globally constant parameters, fixed both in time and space, are shown without superscripts. Most of the direct NN predictions are denoted by the Greek letter $\alpha$, unless

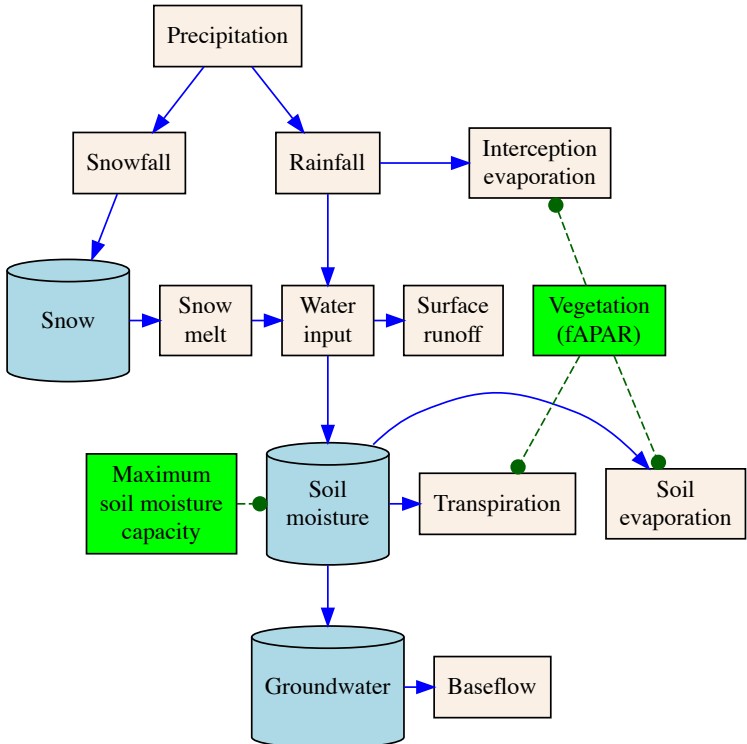

**Figure 1.** Simplified overview of the conceptual hydrological model: beige boxes show water fluxes, blue buckets (cylinders) show water storages, and blue arrows show how water can move from/to water storages. Green boxes show direct predictions of vegetation-related parameters vegetation state (used to partition evapotranspiration into its components) and maximum soil moisture capacity (used to model soil moisture).

the parameter has a clear name and hence, the designated name (e.g., fAPAR). The Greek letter $\beta$ is used to represent globally
constant parameters directly learned by the NN.

The quantified evapotranspiration

$$ET^{<s,t>} = E_i^{<s,t>} + E_s^{<s,t>} + T^{<s,t>} \qquad \left(\text{in mm day}^{-1}\right) \qquad (1)$$

refers to the sum of transpiration, soil and interception evaporation.

The interception evaporation

$$E_i^{<s,t>} = \min\left(\min\left(\text{rainfall}^{<s,t>}, \text{fAPAR}^{<s,t>} \cdot \alpha_{E_i}^{<s,t>}\right), R_n^{<s,t>}\right) \qquad \left(\text{in mm day}^{-1}\right) \qquad (2)$$

is modeled as the amount of water that is intercepted by the vegetation (represented by a flexible scaling of fAPAR), constrained by the amount of rainfall and available energy.

There, fAPAR $(-)$ is the predicted daily vegetation state, $0 < \alpha_{E_i}$ is a direct NN prediction for scaling fAPAR to interception storage capacity, and $R_n$ is available energy expressed as $\left(\text{mm day}^{-1}\right)$ via the latent heat of evaporation.

The modelling of soil evaporation and transpiration

$$E_s^{<s,t>} = \left(1 - \text{fAPAR}^{<s,t>}\right) \cdot ET_{pot}^{<s,t>} \cdot \alpha_{E_s}^{<s,t>} \qquad \left(\text{in mm day}^{-1}\right) \tag{3}$$

$$T^{<s,t>} = \text{fAPAR}^{<s,t>} \cdot ET_{pot}^{<s,t>} \cdot \alpha_T^{<s,t>} \qquad \left(\text{in mm day}^{-1}\right) \quad, \tag{4}$$

respectively, follows traditional, conceptual two-source models where fAPAR partitions the available energy for the soil and plant canopies. The directly predicted parameters by NN, i.e., $\alpha_T$ and $\alpha_{E_s}$, are bounded to the interval $[0,1]$ and represent
effective conductance or 'stress'.

Incoming water

$$w_{\text{in}}^{<s,t>} = \text{rainfall}^{<s,t>} + s_{\text{melt}}^{<s,t>} - E_i^{<s,t>} \qquad \left(\text{in mm day}^{-1}\right) \quad, \tag{5}$$

is distributed to surface runoff, soil moisture and ground water recharge (Appendix A). The relative partitioning among the three water pathways is regulated by the soil moisture state and predictions by the neural network. Soil recharge fraction

$$r_{\text{soil}_{\text{fraction}}}^{<s,t>} = \min\left(1, \left(\frac{SM_{\text{max}}^{<s>} - SM^{<s,t>}}{\max\left(w_{\text{in}}^{<s,t>}, \epsilon\right)}\right)\right) \cdot \alpha_{r_{\text{soil}}}^{<s,t>} \qquad (-) \tag{6}$$

represents the fraction of incoming water that will recharge the soil and scales with the soil moisture deficit relative to the incoming water. There, $0 < SM_{\text{max}}$ (in mm) is the maximum plant available soil water storage capacity and $0 < \alpha_{r_{\text{soil}}} < 1$ represents uncertain processes. Both parameters are directly learned by NN. The additive term $\epsilon = 10^{-8}$ asserts the function is differentiable under all circumstances, which is important for stable NN training.

The groundwater recharge fraction

$$r_{\text{gw}_{\text{fraction}}}^{<s,t>} = \left(1 - r_{soil_{\text{fraction}}}^{<s,t>}\right) \cdot \alpha_{r_{gw}}^{<s,t>} \qquad (-) \tag{7}$$

is modeled as a function of soil recharge fraction and a NN-learned parameter $0 < \alpha_{r_{\text{gw}}} < 1$. The soil recharge fraction and the NN-learned parameter $\alpha_{r_{\text{gw}}}$ are used to model the fraction of surface runoff

$$q_{\text{surf}_{\text{fraction}}}^{<s,t>} = \left(1 - r_{\text{soil}_{\text{fraction}}}^{<s,t>}\right) \cdot \left(1 - \alpha_{r_{\text{gw}}}^{<s,t>}\right) \qquad (-) \quad . \tag{8}$$

The proposed hydrological model equations ensure that the amount of water entering the system (i.e., precipitation) equals the amount of water leaving the system (i.e., evapotranspiration and runoff) plus any change in storage within the system. This constraint ensures that the neural networks also adhere to the principle of mass balance (Appendix A).

### 2.2.2   Dynamic module

Estimations of the parameters that are represented in the dynamic module vary both in space and time. Time-series forcings of
meteorology (net radiation, air temperature and precipitation) at time step $t$, estimated vegetation and water states at time step $t-1$ and compressed representations of the static input are given to an LSTM model as inputs. LSTM is a type of recurrent neural networks (RNN) that is designed to process sequential data (e.g. time-series). Apart from the input mentioned, LSTM

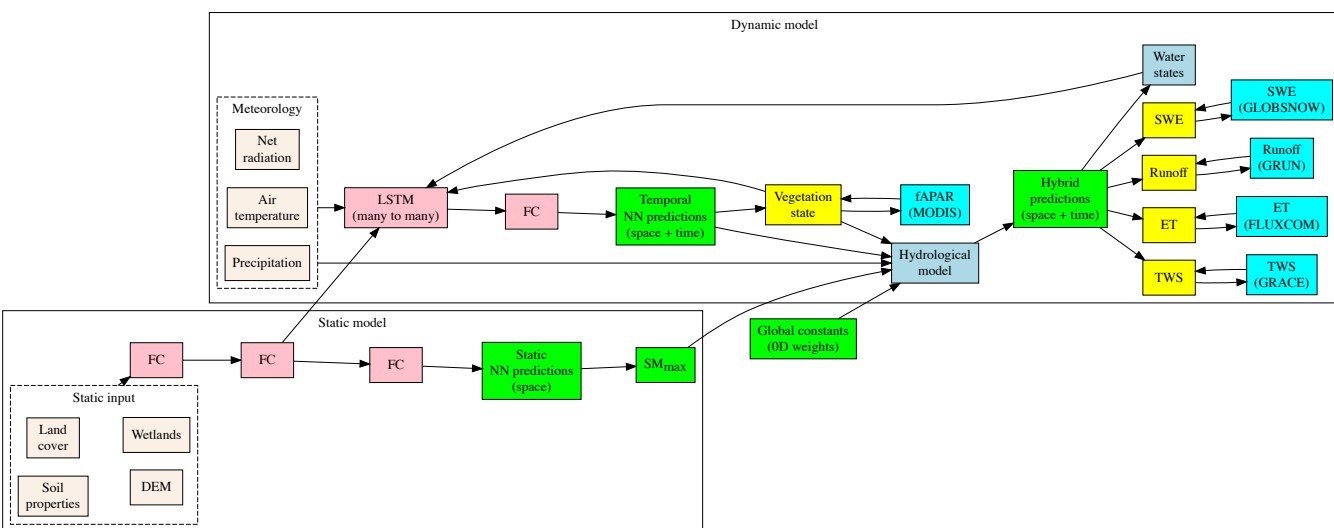

**Figure 2.** High level overview of H2MV: beige boxes show inputs, pink boxes are NN layers, green boxes are predictions, yellow boxes are predictions that are directly constrained and cyan boxes are the corresponding data constraints.

also recieves its own internal hidden and cell states at time step $t-1$, that are responsible for carrying useful information from the previous steps to the prediction of future steps (e.g. memory effect). The output of LSTM is then fed into a fully-connected

(FC) layer (Goodfellow et al., 2016) where they are transformed into interpretable physical parameters. These direct predictions mostly represent the uncertain parameters that are directly connected to a process layer (hydrological cycle) where the process equations occur. The process layer also receives the same time-series forcings of meteorology that are fed into LSTM as inputs. It outputs hybrid (intermediate) predictions, some of which (SWE, runoff, ET and TWS anomalies) are directly constrained using observational data products. Note that the vegetation state (fAPAR) is directly learned and constrained (Fig. 2). The

temporal resolution of the dynamic module is one day, and the spatial resolution is 1°.

### 2.2.3 Static module

In the static module, static features representing land-surface characteristics are fed into a FC layer that is connected to another FC layer. The first FC layer represents higher-dimensional patterns of the original input, while the second FC layer reduces (compresses) the higher-dimensional representation. The compressed data are then given to the LSTM layer (dynamic module)

and connected to a final FC layer. The last FC layer is responsible for transforming the compressed representation of the static features into an interpretable and spatially varying hydrological parameter ($SM_{max}$), that is connected to the process layer (hydrological cycle) in the dynamic module. Note that the static module is explicitly connected to the dynamic module in two ways: connection between the output of FC layer to LSTM and the connection between the spatially varying estimation and process layer (Fig. 2). There is also implicit connection between the two sub-modules as the full model is trained end-to-end

and during optimization learned spatially-varying parameters are updated in order to minimize the loss.

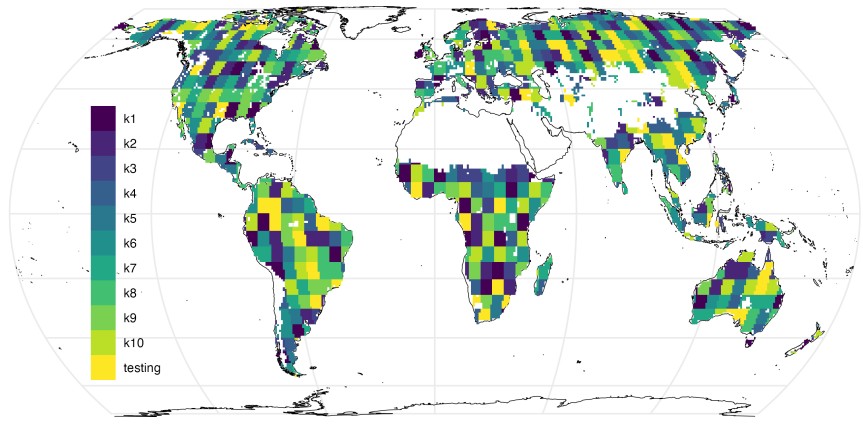

**Figure 3.** Validation sets for 10 different models and a fixed testing set. Note that, during training, each fold has a separate and unique validation set and all models were tested on the same testing set.

Global constants (fixed both in space and time) are trainable parameters that are not directly connected to the input (Fig. 2). During model optimization these parameters are updated. This means the input and the constraints have an indirect impact on these learned parameters.

## 2.3 Model optimization

### 2.3.1 Cross-validation

We employ a 10-fold cross-validation (CV) to train and validate H2MV, which entails training 10 separate models, each with distinct training and validation sets. Additionally, the weights of each model are randomly initialized during training. The objectives of the CV are twofold: to evaluate the generalization capability of the model and to gain insights into the equifinality of model estimations.

To mitigate spatial autocorrelation, we implement spatial blocking as suggested by Roberts et al. (2017). During the training of each fold, a unique set of validation data is utilized to validate the model. It is important to note that a separate testing dataset, unseen by any of the models during training, is used to assess the model's performance, robustness, and equifinality after all models are trained (Fig. 3).

### 2.3.2 Loss function

To quantify the performance of the hybrid model for any input data $X$, NN weights $\Theta$, and global constraints $\beta$, we use the mean squared error (MSE)

$$L\left(X, \Theta, \beta\right) = \frac{1}{N_c} \sum_{c=1}^{C} \sum_{i=1}^{N_c} (y_{c,i} - \hat{y}_{c,i})^2 \tag{9}$$

as a loss function that aggregates individual losses to obtain a final loss term. Here, $C$ is the number of data constraints, $N_c$ is the number of examples (data points) in the constraint $c$, and $y_{c,i}$ and $\hat{y}_{c,i}$ are the observed and predicted values of the data constraint $c$, respectively. During training, $\Theta$ and $\beta$ are updated to minimize the total loss $L$.

We apply a Z-transformation to both the predictions and their corresponding observations before calculating the loss. This addresses the issue of differing units among model constraints and ensures that each individual loss has a similar impact on the model's behavior.

### 2.3.3 Model Training

We use Z-transformation to standardize both inputs and outputs (targets) of H2MV during training. We use the unscaled forcing data to compute hydrological equations, ensuring proper constraint of the water balance. For optimization, we opt for the Adam optimizer (Kingma and Ba, 2014). During optimization, the learnable parameters (e.g., weights) of both the dynamic and static NN are updated to minimize the total loss. To prevent overfitting, early stopping is implemented, halting the training process once the model's performance on the validation set ceases to improve. Additionally, we run the full model without updating weights to stabilize water and vegetation states (spin-up), which are then fed as inputs to the LSTM network at each iteration during training. The model with the smallest total loss on the validation set during training is selected as the final model. This best-performing model is then utilized to make the final predictions on the testing set.

## 2.4 Model evaluation

### 2.4.1 Performance metrics

We evaluate our model's performance using root mean square error (RMSE), Pearson's correlation coefficient (r), and the standard deviation ratio (SDR), which is the ratio between the predicted and observed standard deviations.

### 2.4.2 Mean seasonal cycle

We define the mean seasonal cycle (MSC) as follows:

$$MSC\left(m\right) = \frac{1}{Y} \sum_{y=1}^{Y} p_{m,y} \tag{10}$$

where $p_{m,y}$ represents the modeled or observed parameter for month $m$ and year $y$, and $Y$ is the total number of years.

### 2.4.3 Interannual variability

We define the interannual variability (IAV) as follows:

$$IAV(m, y) = p_{m,y} - MSC(m) \tag{11}$$

In this equation, $p_{m,y}$ denotes the modeled or observed parameter for a given month $m$ and year $y$, while $MSC(m)$ represents the mean seasonal cycle for that specific month $m$.

### 2.4.4 Equifinality evaluation

In H2MV, we incorporate a relatively high number of processes while being constrained by a limited set of observational data. This makes H2MV susceptible to equifinality. To address this, we use a 10-fold CV method, training 10 models with varying sets of training and validation data, and initializing each model's weights randomly.

This approach allows us to evaluate the sensitivity of parameter estimations to three key factors: 1) the validation set, 2) initial NN weights, or 3) the combination of both. If we observe considerable variability in the parameter estimations among the 10 trained models, it suggests that the estimations for a particular parameter are equifinal. This means that the parameter is subject to high uncertainty, as multiple mechanisms within the model can lead to similar outcomes. In essence, our analysis of equifinality helps determine whether a simulation of a variable in the model, particularly fluxes and states, are under-constrained by the observational and theoretical constraints we have applied.

We use a single, normalized metric value for each estimated parameter across the 10 models, facilitating a clearer understanding of the level of equifinality in the estimations. This metric represents the average error between different model realizations and therefore represents the variability of a certain parameter.

Following Gupta et al. (2009), we use the decomposition of MSE into phase, bias, and variance errors

$$e_{\text{phase}} = \frac{1}{N_p} \cdot \sum_{p=1}^{N_p} \frac{2 \cdot \sigma_{p,1} \cdot \sigma_{p,2} \cdot (1 - r_p)}{v_p} \tag{12}$$

$$e_{\text{bias}} = \frac{1}{N_p} \cdot \sum_{p=1}^{N_p} \frac{(\mu_{p,1} - \mu_{p,1})^2}{v_p} \tag{13}$$

$$e_{\text{var}} = \frac{1}{N_p} \cdot \sum_{p=1}^{N_p} \frac{(\sigma_{p,1} - \sigma_{p,1})^2}{v_p} \quad , \tag{14}$$

respectively. Here, $p$ represents a pair of estimations for the same parameter obtained from two different models through CV, $N_p$ is the total number of such pairs, and $\sigma_{p,1}$ and $\sigma_{p,2}$ denote the standard deviations of the first and second estimations in the pair $p$ respectively. Further, $r_p$ represents the correlation between the first and second estimations, while $v_p$ is the mean variance between these estimations. Additionally, $\mu_{p,1}$ and $\mu_{p,2}$ denote the mean of the first and second estimations in the pair $p$, respectively. We normalize all of these error terms by the mean variance between the two estimations to account for different units. The computation is performed exclusively on the predictions from the testing set.

We define equifinality index (EI) as follows:

$$EI = e_{\text{phase}} + e_{\text{bias}} + e_{\text{var}} \tag{15}$$

EI is essentially MSE normalised by variance of the estimations. Higher EI values signify a larger degree of equifinality, or reduced robustness, while lower values indicate smaller equifinality, and therefore a more robust prediction.

### 2.4.5 TWS decomposition

In our model, TWS is composed of three primary water storage components that are not directly observed, except for snow water equivalent, which is derived from observational data. It is crucial and intriguing to evaluate which component of TWS is most dominant and where this dominance occurs spatially. This is particularly important because previous studies have highlighted significant modeling uncertainties related to these components (Trautmann et al., 2018; Kraft et al., 2022).

To decompose TWS variability we use a technique introduced by Getirana et al. (2017).

First, we compute the absolute contribution for each water storage

$$C_{\text{abs}}(S) = \sum_{t=1}^{T} \left| S_t - \overline{S} \right| \quad , \tag{16}$$

with $T$ being the total number of time steps, $S_t$ the water storage at the time step $1 < t < T$, and $\overline{S}$ the mean of the water storage $S$ over time. The relative contribution of each modeled water storage

$$C_{\text{rel}}(S_i) = \frac{C_{\text{abs}}(S_i)}{\sum_{S_j=1}^{N_S} C_{\text{abs},S_j}} \in [0,1] \tag{17}$$

is then defined for all modeled water storages $N_S$.

## 3 Results and discussion

### 3.1 Model performance

Here the model evaluation is done on the same independent test set for all the members (Fig. 3); this data has not been seen during model training. The additional experiment where we test our model's spatio-temporal generalization capability is discussed in the Appendix C.

On the global scale, the observed patterns of fAPAR are well-reproduced and robust across CV members (Fig. 4a). The MSC of fAPAR is well-captured, although there is some disagreement between the predictions and observations in December (Fig. 4b), possibly due to artefacts in the satellite based fAPAR product due to snow contamination. The IAV, in contrast, is more challenging to predict and the agreement with the observations is lower. While the general dynamics of the IAV are represented relatively well, the trend is not reproduced by the model (Fig. 4c). The model also captures the observed patterns of fAPAR for all major regions (Fig. B4).

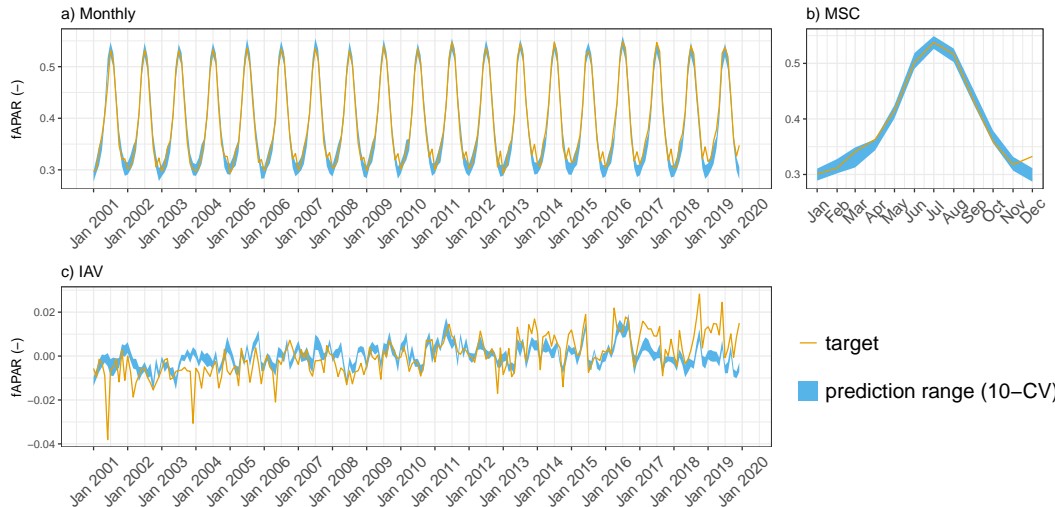

**Figure 4.** Predicted versus observed mean fAPAR over the testing set (spatial domain) across different folds: a) monthly, b) mean seasonal cycle and c) interannual variability.

The TWS is well-reproduced on the global scale (Fig. 5a). The MSC matches the observations in terms of dynamics and timing (Fig. 5b). There is a slight phase shift and underestimation in the amplitude of the TWS predictions. A similar pattern was noticed in previous studies (Kraft et al., 2022; Trautmann et al., 2022) and is likely related to the missing representation of surface-water variations with snow melt in the northern hemisphere. Figure 5c shows that the patterns of TWS IAV are captured well between 2002 and 2014, while there is a shift afterwards. Overall, TWS predictions are robust across the CV members.

H2MV reproduces patterns of SWE, ET, and runoff well. We show the model performance on these data in the Appendix B1. The ET and runoff are reproduced well on all temporal scales (Fig. B2 and B3). These variables have been upscaled from sparse observations using ML, and hence, they are not directly observed. We do, therefore, expect H2MV to be able to represent these variables well. The SWE, in contrast, is directly observed. Here, the model represents the IAV relatively well, but the MSC amplitude is underestimated (Fig. B1). The underestimation of SWE could be linked to the lack of representation of surface water storage. To reduce the TWS phase shift, the model may need to reduce snow accumulation, as it has no mechanism to buffer the melt water. Furthermore, additional mass accumulation via snow in the high latitudes would lead to a larger error in TWS, which already matches the observations well from January to March. Similarly, larger SWE would lead to an increased runoff in northern spring, increasing the runoff error. Hence, the low SWE may be caused by various trade-offs, and inconsistencies among data streams including precipitation, which is very uncertain with respect to snowfall.

Overall, the seasonality has been reproduced well for all target variables in terms of Pearson correlation ($r$) with values close to 1, while the correlation varied for the interannual variability (IAV), ranging from 0.47 to 0.83 (Fig. 6). In terms of the RMSE, IAV generally shows lower RMSEs, except for TWS. The SDR (the ratio between predicted and observed standard

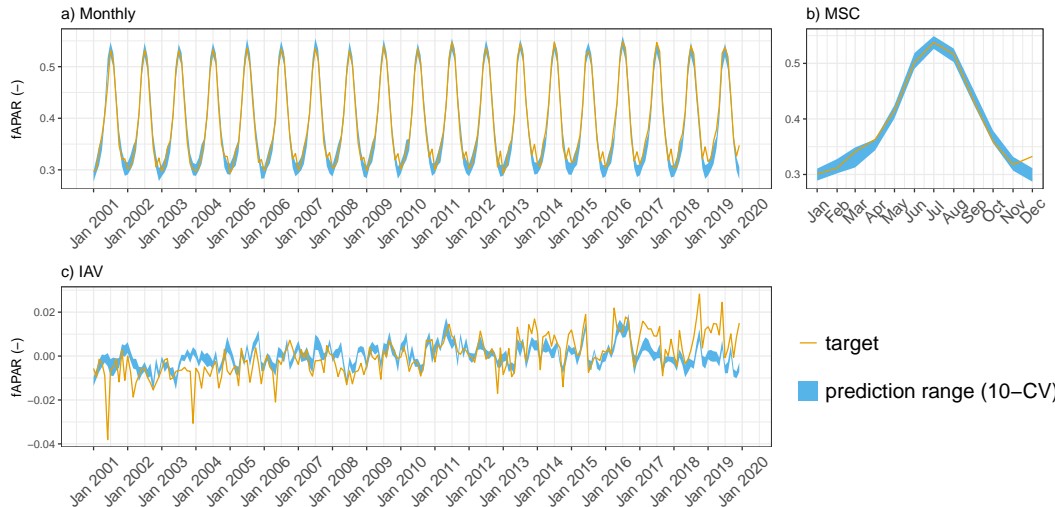

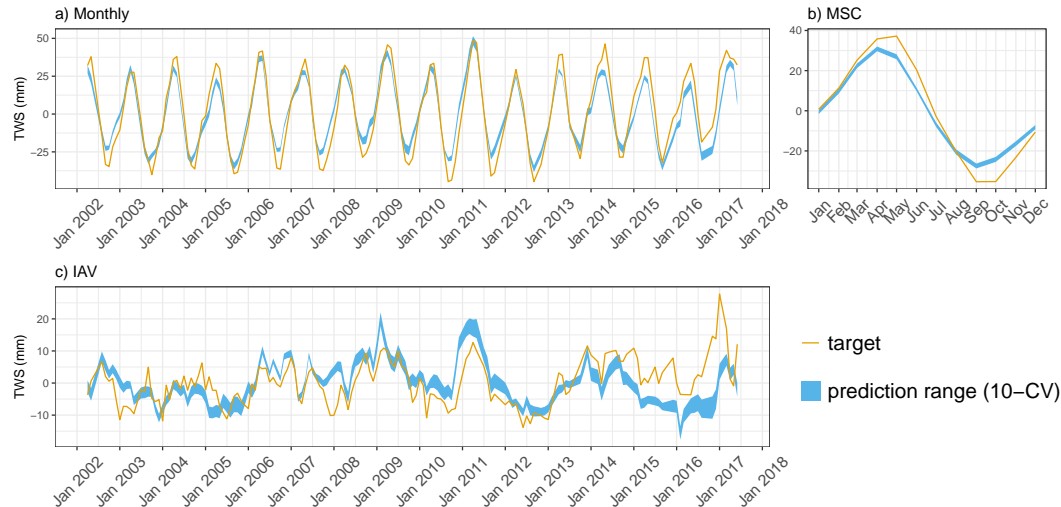

**Figure 5.** Predicted versus observed mean TWS (anomaly) over the testing set (spatial domain) across different folds: a) monthly, b) mean seasonal cycle and c) interannual variability.

deviation) indicates that fAPAR seasonality is well represented by the model in terms of variability, while the IAV magnitude is underestimated. The TWS variability is underestimated due to an underestimation of seasonal amplitude, while the interannual variance is matched well. The SWE is underestimated with an SDR of 0.75 for the mean seasonal cycle (MSC) and 0.5 for the IAV. Both ET and runoff are matched well in terms of variance, expect for the ET IAV, which is overestimated by a factor of two. The apparent overestimation of ET interannual variance by the model is likely due to a substantial underestimation of interannual variance by the FLUXCOM approach (Jung et al., 2019) used to generate the reference ET product.

Overall, H2MV performance is qualitatively consistent with the findings of Kraft et al. (2022). For a comprehensive assessment of the model's global performance and comparison with the results from Kraft et al. (2022), refer to Fig. B5.

### 3.2 Equifinality of the intermediate predictions

Here, we assess the equifinality of H2MV's predictions regarding water states, as illustrated in Figure 7. Figure 7a displays the predicted anomalies of each modeled water state across different models (represented by the thickness of the lines that shows the range of the estimations). Predicted anomaly refers to the predicted state minus the mean of the predicted state. Notably, the dynamic patterns of all modeled water storages exhibit high robustness, indicating that temporal patterns are neither sensitive to the random weight initialization of the neural network during training, nor to the different training/validation set splits (Fig. 7a).

However, upon assessing the means of the trained models, it becomes evident that there is uncertainty regarding the mean values of the water storages (Fig. 7b), particularly for SM and TWS. It is worth noting that SWE is well constrained, which is expected as it is directly constrained by the observational data in high latitudes. Figure 7c illustrates a positive correlation

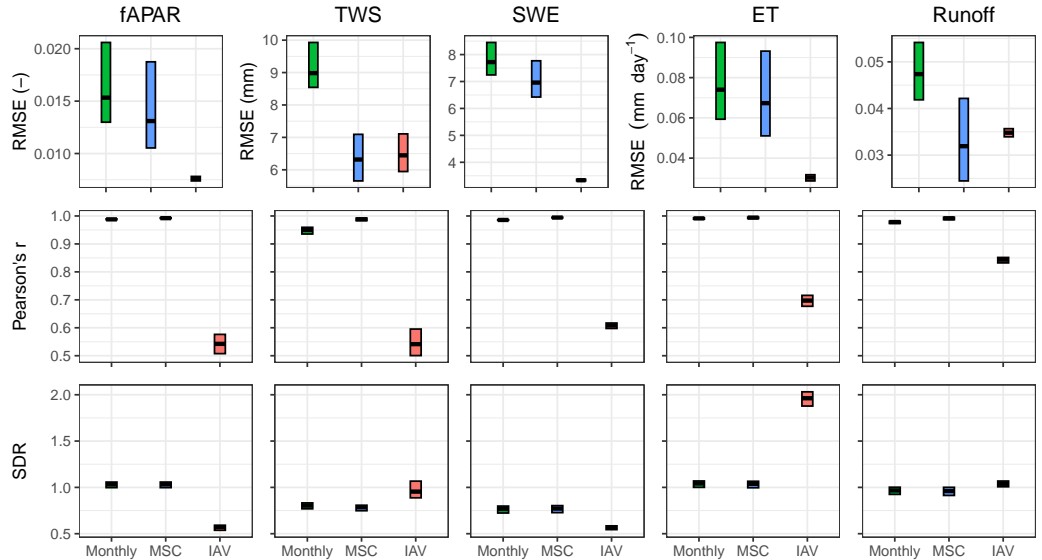

**Figure 6.** Model performance on the testing set. Cross bars show the maximum and minimum error, and the lines show the mean error across 10 folds. The rows are metrics and the columns are model constraints. RMSE refers to root mean squared error, and SDR is standard deviation ratio (the ratio between predicted and observed standard deviation).

between the predicted mean of SM and TWS. The source of this variability of the means of SM and TWS may be caused by the uncertainty in estimating the magnitude of $SM_{\max}$ (note that the estimated spatial patterns of $SM_{\max}$ are very robust), which

325 provides the upper bound of the soil moisture water storage. Therefore, by constraining the estimations of $SM_{\max}$, we could potentially improve SM predictions. Given that TWS is the sum of SM, GW, and SWE, and since SWE is already robustly estimated, constraining $SM_{\max}$ would consequently provide a more robust estimate of GW and TWS. Interestingly, the small uncertainty observed in the mean value of GW (Fig. 7b) does not appear to be related to either the uncertainty in SM or TWS (Fig. 7c). Enhancing H2MV's representations of the parameters that control GW dynamics could improve our representation of

330 SM and, consequently, TWS. This suggests that by choosing to refine our constraints on either GW or SM—whether through incorporating more process details or applying data constraints—we could indirectly improve our estimates of the other water state as well, thus presenting another promising avenue for future work.

The analysis of the equifinality reveals that, overall, the most dominant error component of MSE in H2MV is phase shift (covariance error) (Fig. 8). This could be attributed to the fact that most of H2MV constraints operate at a monthly temporal

335 resolution, whereas H2MV operates at a daily temporal resolution at which the metric was calculated. At the same time, a phase shift may occur due to missing representation of surface water storage and river routing.

Equifinality metric values for $SM_{\max}$, SM, and TWS (Fig. 8a), as well as soil and interception evaporation (Fig. 8b), groundwater recharge, predicted fractions (Fig. 8d), and snow melt (Fig. 8e), are relatively large. In contrast, the remaining parameters exhibit relatively small equifinality values, all being smaller than 0.1.

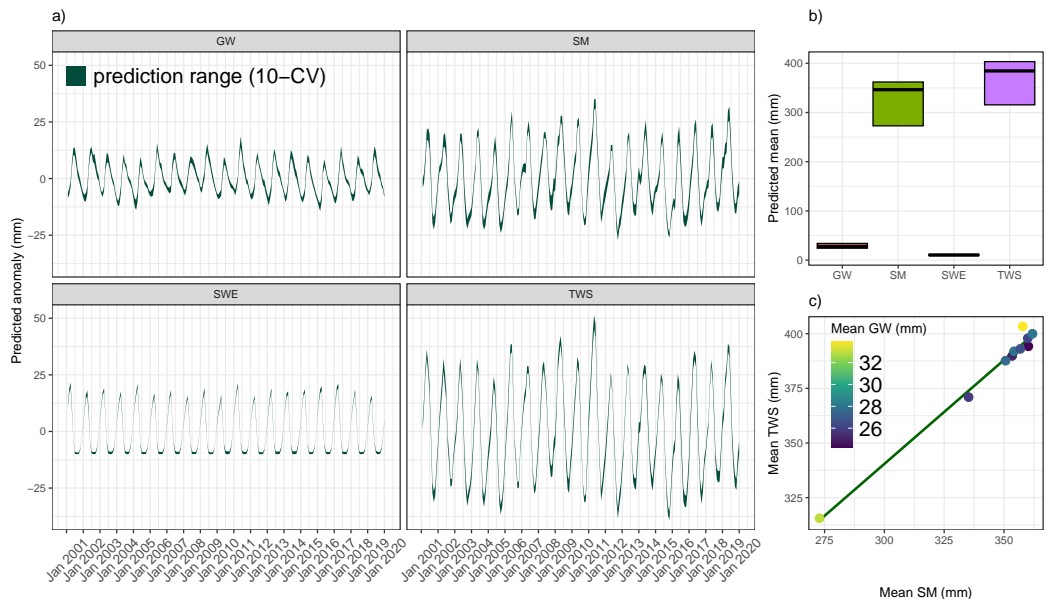

**Figure 7.** Predicted water states averaged over testing set across 10 different folds: the thickness of the line show the range of the estimations. a) Predicted anomalies (state - mean(state)), b) Range of means across folds: the lines show the average mean and the cross-bars show the maximum and minimum mean values across the folds, c) Predicted means of SM versus TWS: the points are different folds and the line is the regression line. Colors of the points indicate the values of GW.

GW demonstrates a smaller equifinality value relative to TWS and SM (Fig. 8a), which supports the information depicted in Fig. 7. Notably, transpiration is predicted more robustly compared to interception and soil evaporation (Fig. 8b), indicating that equifinality of ET partitioning is primarily between soil and interception evaporation. This indicates that the identifiability of the neural network's learned parameters for evapotranspiration partitioning is weak, particularly for the parameters used to model soil and interception evaporation.

Snow accumulation is highly robust, primarily governed by represented processes with limited impact from the NN (due to globally constant snow correction) (Fig. 8e).

## 3.3 Emerging global patterns

One of the capabilities of the proposed hybrid model is to retrieve information on intermediate processes and patterns that lack direct observational constraints. This section presents some of the emerging global patterns after training the model.

### 3.3.1 Evaporative Fraction

The Evaporative Fraction (EF), defined as the ratio of evapotranspiration to the total available energy (net radiation), serves as a valuable intermediate parameter shedding light on whether the Earth's surface is dominated by evaporation (in areas

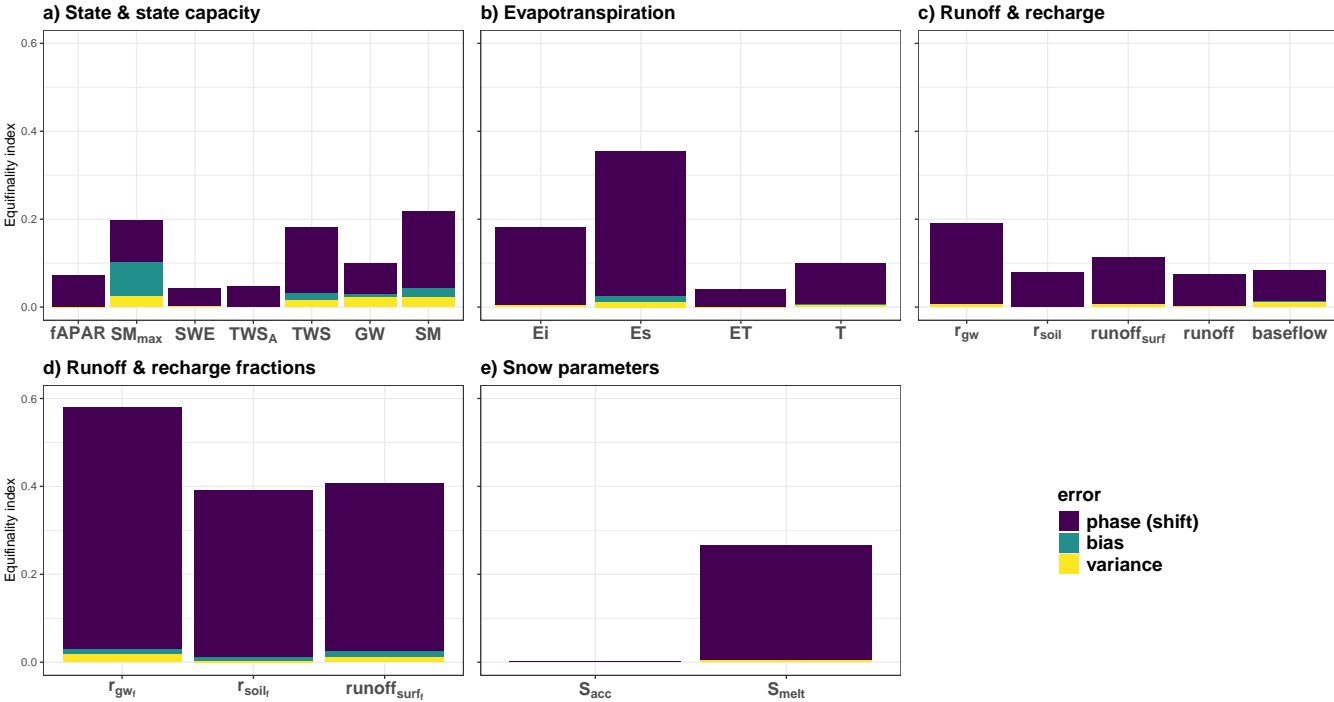

**Figure 8.** Equifinality index averaged over all the combinations of 10 folds: $TWS_A$ indicates anomalies of TWS, $E_i$ is interception evaporation, $E_s$ is soil evaporation, ET is evapotranspiration, T is transpiration, $r_{gw}$ refers to groundwater recharge, $r_{soil}$ is soil recharge, $runoff_{surf}$ is surface runoff, the subscript f refers to fractions of these components, $s_{acc}$ and $s_{melt}$ are snow accumulation and melt, respectively.

with ample water availability) or sensible heat flux (in water-limited regions). As depicted in Fig. 9a, higher EF values are anticipated predominantly in the southeast of North America, much of Central and South America, Western Europe, Central
355 Africa, and Southeast Asia. These regions typically experience moderate to high precipitation levels and boast significant vegetation coverage. Conversely, relatively low EF values are projected for most of Canada and the southwestern United States (US), specific eastern regions of Brazil, the southwestern part of South America, extensive areas of Western Russia, the southern and western regions of Africa, and most of Australia. It is worth noting that this result is based on the predicted ET that is constrained using observation-based data, and net radiation which is a meteorological input to the model.

360 **3.3.2 Runoff coefficient**

The runoff coefficient, representing the ratio of total runoff to precipitation, serves as a critical indicator of how much precipitation transforms into runoff rather than being absorbed into the soil, evaporated, or transpired by vegetation. H2MV projects varying runoff coefficient values across different regions. Moderate to high values are anticipated for the Northeast and Northwest of North America, the Amazon basin, much of the northern part of South America, Northern Europe, extensive areas of

Russia, Southeast Asia, and New Zealand. Conversely, low runoff coefficient values are forecasted for central and southern regions of North America, specific eastern areas of Brazil, most of the southwestern part of South America, parts of Central Asia, and Australia (see Fig. 9b). This outcome strongly aligns with global trends identified in a comprehensive study by Wang et al. (2022), which analyzed data from 23 advanced models within the Coupled Model Intercomparison Project Phase 6 (CMIP6). This result is derived from model's constrained estimation of runoff and precipitation that is one of the key meteorological inputs of the model.

### 3.3.3 Transpiration versus evapotranspiration

The ratio of transpiration to evapotranspiration reflects the amount of water transpired by the vegetation relative to the total water leaving the surface. Transpiration is very important for both understanding water cycle components and the coupling between carbon and water cycles. Figure 9c reveals that, globally in most places, transpiration is the more dominant parameter compared to the other modeled components (interception and soil evaporation) of ET. Specifically, the highest domination of transpiration can be seen in northwest and southeast of Canada, most parts of South America (especially the Amazon basin area), high latitudes of Europe and Asia, and Congo basin in Central Africa. These regions are known to have moderate to high amount of vegetation with moderate to high annual precipitation patterns. Most of the low values were predicted to be around arid regions, that are known to have low amount of vegetation. Overall, our findings (mainly spatial patterns) align qualitatively with reported estimations by Martens et al. (2017), Wei et al. (2017) and Nelson et al. (2024). However, compared to these findings, H2MV indicates a more pronounced dominance of transpiration in the Amazon and Congo basins compared to other regions within their respective continents. Note that this comparison focuses on spatial patterns rather than on magnitudes.

### 3.3.4 Maximum soil moisture content

The maximum soil moisture content available for plant transpiration, denoted as $SM_{max}$ (also known as rooting zone water-storage capacity), represents a crucial parameter in climate modeling, particularly for studying carbon-water cycle processes. However, our current grasp of this parameter, especially its spatial variability, remains highly limited due to the lack of direct observations. Several studies (Wang-Erlandsson et al., 2016; Tian et al., 2019; Stocker et al., 2023), as well as related research on plant rooting depth (Yang et al., 2016; Fan et al., 2017), have attempted to estimate this parameter. While there are qualitative agreements among these studies, significant discrepancies exist, likely stemming from diverse methodologies and underlying assumptions. A noteworthy aspect of our proposed model is its direct learning of $SM_{max}$ from static inputs (such as land cover and soil properties) using neural networks. Globally, H2MV predicts high spatial variability for $SM_{max}$ (Fig. 9d). The highest $SM_{max}$ values are predominantly estimated in South America, Central Africa, Southeast Asia, and the extreme northern and southern regions of Australia. This observation aligns with the regions known for substantial and seasonal rainfall, abundant radiation, and extensive vegetation coverage. Conversely, the lowest $SM_{max}$ values are identified in the high latitudes of the Northern Hemisphere. Interestingly, there are substantial qualitative agreements, in terms of spatial patterns, between our estimations and those reported by Wang-Erlandsson et al. (2016), Tian et al. (2019), and Stocker et al. (2023). For instance, these studies, along with our own, predict higher values across much of South America, Central Africa, and Southeast Asia.

Conversely, they estimate significantly lower values for the high latitudes of the Northern Hemisphere. Our estimations are more closely aligned, in terms of magnitude, with those reported by Stocker et al. (2023). In contrast, both Wang-Erlandsson et al. (2016) and Tian et al. (2019) report significantly lower values for this parameter. This discrepancy across different models highlights the necessity for additional global-scale studies and validation efforts concerning this parameter.

### 3.3.5 TWS decomposition

Another critical yet uncertain aspect in hydrological modeling pertains to the contribution of water storages to observed TWS variability. Figure 9e illustrates the breakdown of modeled daily TWS variability into its components, highlighting their relative contributions to TWS variability. In regions of very high latitudes in the Northern Hemisphere, SWE emerges as the dominant factor influencing TWS variability, a finding consistent with existing literature, including studies by Kraft et al. (2022) and Trautmann et al. (2022). Conversely, the contribution of GW predominates only in the northwest of South America, a relatively small area in Central Africa (around the Congo basin), and some parts of Southeast Asia. The remainder of terrestrial land globally is estimated to be primarily influenced by SM variability. This finding closely aligns with previous research, particularly that of Kraft et al. (2022), which used similar techniques and datasets.

### 3.3.6 Baseflow index

The Baseflow Index (BFI), indicating the ratio of baseflow to total runoff, plays a crucial role in understanding the proportion of streamflow contributed by baseflow which is discharged from groundwater storage. H2MV's estimations (Fig. 9f) indicate a significant predominance of baseflow in the central regions of North America, Europe, Western Asia, and the Amazon Basin. Conversely, the contribution of baseflow is relatively low in other areas. This estimation is qualitatively consistent with the findings reported in the studies by Beck et al. (2013) and Beck et al. (2015). These studies' and our results show higher BFI values for the mid and high latitudes of North America, the majority of Europe and Western Asia, and regions within South America, particularly the Amazon basin. However, in contrast to these studies, our estimated BFI values for Central Africa are significantly lower.

## 3.4 Challenges and future perspective

H2MV heavily relies on the quality of both input and observed target data, as they directly influence the results. The satellite-based observational data used for model optimization can contain measurement errors. For instance, TWS anomaly (GRACE) (Landerer and Swenson, 2012; Soltani et al., 2021), fAPAR (MODIS) (Xu et al., 2018), and SWE (GLOBSNOW) (Luojus et al., 2021) are known to exhibit significant uncertainties. Furthermore, both runoff and ET products are not directly observed on a global scale, thus are expected to have significant uncertainties (Ghiggi et al., 2019; Jung et al., 2019). The total uncertainty, which includes the uncertainty in the input data, may substantially impact the estimations of the represented parameters. However, it is important to note that hybrid modeling may be less sensitive to the uncertainty in the target data compared to a purely data-driven approach, such as pure ML, due to the incorporation of process knowledge that governs the predictions to

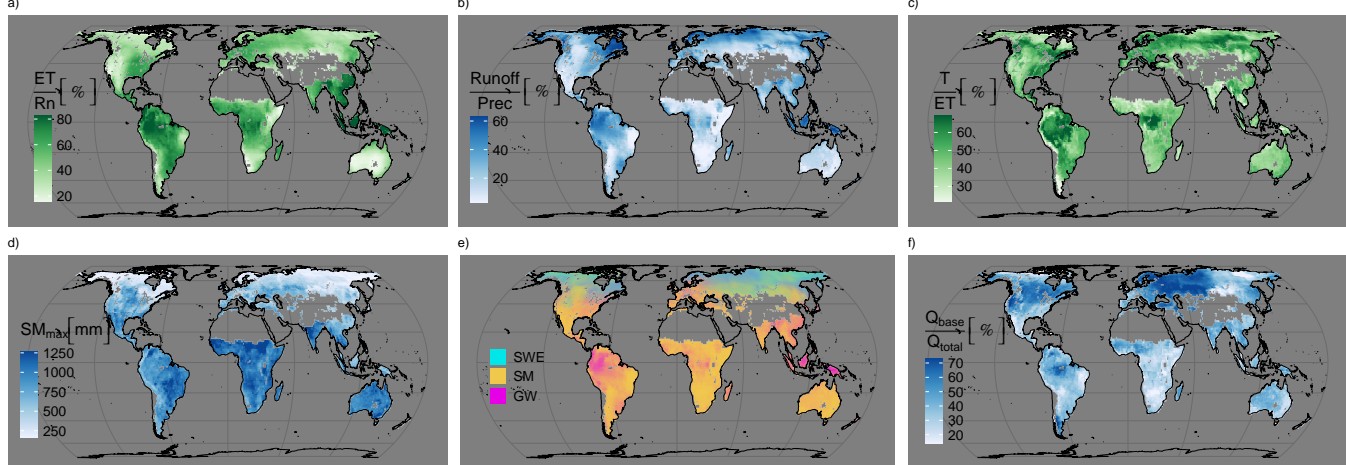

**Figure 9.** Emerging intermediate global patterns averaged across 10 folds: a) the ratio of evapotranspiration to total net radiation (evaporative fraction), b) the ratio of runoff to precipitation (runoff coefficient), c) the ratio of transpiration to evapotranspiration, d) predicted maximum soil moisture capacity (rooting zone water-storage capacity), e) decomposition of terrestrial water storage into snow water equivalent (SWE), soil moisture (SM) and groundwater storage (GW), f) the ratio of baseflow ($Q_{base}$) to the total runoff ($Q_{total}$) (baseflow index).

some extent. For instance, we calibrate our estimates of ET using the FLUXCOM ET product as a benchmark. Upon comparing
the IAV of our ET estimates with FLUXCOM data, it becomes evident that H2MV tends to overestimate IAV. This discrepancy
is actually plausible, considering that FLUXCOM is known to substantially underestimate ET's IAV (Jung et al., 2019).

Another challenge arises in balancing the estimation of more uncertain processes with their interpretability. Currently, we
have a limited number of constraints for the modeled hydrological components. Adding more processes to the model without
incorporating additional data constraints is likely to introduce more equifinality, unless the implementation of the process
requires no or few parameters to calibrate. Despite its apparent simplicity, H2MV represents a relatively high number of
water cycle processes from a hybrid modeling perspective. However, the directly learned uncertain parameter estimations by
the NN should be interpreted with caution. As a concrete example, we partition ET into transpiration, soil, and interception
evaporation using relatively well-understood processes (e.g., as a function of vegetation and available radiation), along with
uncertain parameters directly learned by the NN. We directly predict three parameters (one for each component of ET), and
theoretically, there could be infinite combinations of these parameters that can lead to the same ET value (equifinality). While
our method to assess equifinality provides valuable insights into the robustness of our estimations, it does not guarantee that
parameters with very high robustness across 10 different models with different weight initializations (in a 10-fold CV setup)
are not equifinal. This is because we do not explore the weight space to its full extent, and there are many hyperparameters of
the NN that can impact the robustness of our predictions.

One of our next objectives is to delve deeper into understanding the uncertainty surrounding the mean estimation of SM, which appears to correlate with the mean of TWS. Investigating whether refining and constraining $SM_{\mathrm{max}}$ estimation leads to a more accurate representation of SM, GW and TWS would be particularly intriguing.

    Furthermore, our approach enables coupling the hydrological model with the carbon cycle. This coupling could substantially enhance our understanding of both the water and carbon cycles, as well as their interactions. By incorporating additional

observational satellite data products related to the carbon cycle, we can further elucidate these complex interactions. Given that H2MV already represents important carbon cycle-related parameters such as vegetation state, $SM_{\mathrm{max}}$, and transpiration, it provides a unique avenue for studying key water-carbon cycle interactions that remain largely uncertain in current research (Humphrey et al., 2018; Jung et al., 2017; Gentine et al., 2019).

## 4   Conclusions

This study delves into the concept of combining machine learning with process knowledge to model the global terrestrial hydrological cycle. The proposed hybrid model learns physically interpretable parameters, coefficient and variables from input meteorology and static land features. These learned parameters are then seamlessly integrated into a process layer where computations of the hydrological cycle occur.

    A key innovation of the proposed model lies in its explicit learning of vegetation-related state parameters, which have been

shown to directly influence the water cycle but are not commonly utilized in hydrological modeling. These parameters include fAPAR, constrained against satellite observations, and maximum soil moisture capacity, directly learned from the static land features.

    During model evaluation against observations, we find a high overall agreement between the predictions and the observed data. Additionally, we assess the learned global patterns of several intermediate hydrological parameters and find that these

patterns align well with current knowledge.

    Given the inherent flexibility of combining a machine learning model with a process-based model, equifinality is a pivotal challenge. With the quantification of equifinality via CV ensemble uncertainty, we illustrated a pathway to improve hybrid models and to assess their physical consistency. Given the significant flexibility of neural networks, it is important to assess the equifinality of hybrid models (Acuña Espinoza et al., 2023). However, the quantification of equifinality in hybrid models

is often less emphasized in the current literature. We observe that the temporal patterns of the modeled mean global water storages demonstrate high robustness. However, we note that the predicted means of soil moisture and terrestrial water storage lack robustness, indicating equifinality issues within the hybrid model. The covariation observed between the predicted means of soil moisture and terrestrial water storage suggests that refining or constraining $SM_{\mathrm{max}}$ in the model could enhance the representation of soil moisture, groundwater and terrestrial water storage.

*Code and data availability.* The model simulations, aggregated to a monthly resolution, are accessible via DOI: doi.org/10.5281/zenodo.12583615 (Baghirov et al., 2024). The initial release of the complete model code can be accessed at DOI: doi.org/10.5281/zenodo.12608916 (Baghirov, 2024). For the most current version of the code, please visit the public repository at https://github.com/zavud/h2mv. We are open to sharing the original daily simulations and additional variables (that are not shared) upon request.

## Appendix A: Hydrological model

**A1 Snow**

Snow accumulation (snowfall) $(\mathrm{mm\,day^{-1}})$ is a function of air temperature $(T_{\mathrm{air}})$ in $°\mathrm{C}$, and precipitation (prec in $\mathrm{mm\,day^{-1}}$):

$$
s_{acc}^{<s,\,t>} = \begin{cases} prec^{<s,\,t>} \cdot \beta_{snow}, \text{ if } T_{air}^{<s,\,t>} \leq 0°C \\ 0, \text{otherwise} \end{cases} \tag{A1}
$$

Here (Eq.

eqrefeq:sacc-a), $\beta_{\mathrm{snow}}$ is a NN learned parameter (globally constant and $0 < \beta_{\mathrm{snow}} < 1$) that is used to account for the reported

overcorrection of snow (Decharme and Douville, 2006).

We use a degree-day method to model melting of the snow $(\mathrm{mm\,day^{-1}})$:

$$
s_{melt}^{<s,\,t>} = \min\left(\max\left(T_{air}^{<s,\,t>},\,0\right) \cdot \alpha_{s_{melt}}^{<s,\,t>},\, SWE^{<s,\,t-1>}\right) \tag{A2}
$$

where $\alpha_{\mathrm{smelt}}$ $(>0)$ is directly learned by NN. The snow storage snow water equivalent (SWE in $\mathrm{mm}$) is updated as follows:

$$
SWE^{<s,\,t>} = \max\left(SWE^{<s,\,t-1>} + s_{acc}^{<s,\,t>} - s_{melt}^{<s,\,t>},\, 0\right) \tag{A3}
$$

**A2 Evapotranspiration**

Rainfall $(\mathrm{mm\,day^{-1}})$ is simply the total precipitation depending on the temperature:

$$
\text{rainfall}^{<s,\,t>} = \begin{cases} prec^{<s,\,t>}, \text{ if } T_{air}^{<s,\,t>} > 0°C \\ 0, \text{otherwise} \end{cases} \tag{A4}
$$

Interception evaporation ($E_{\mathrm{i}}$ in $\mathrm{mm\,day^{-1}}$) is modeled as the amount of water that is intercepted by the vegetation and that will eventually evaporate back to the atmosphere:

$$
E_i^{<s,\,t>} = \min\left(\min\left(\text{rainfall}^{<s,\,t>},\, \text{fPAR}^{<s,\,t>} \cdot \alpha_{E_i}^{<s,\,t>}\right),\, R_n^{<s,\,t>}\right) \tag{A5}
$$

where, fAPAR (-) is the predicted daily vegetation state, $\alpha_{Ei}$ (> 0) is a direct NN prediction that accounts for uncertain processes, and $R_n$ is net radiation $(\mathrm{mm\,day}^{-1})$. To conserve the energy balance, the net radiation is updated as follows:

$$R_n^{<s,\,t>} = R_n^{<s,\,t>} - E_i^{<s,\,t>} \tag{A6}$$

Potential evapotranspiration ($ET_{\mathrm{pot}}$ in $\mathrm{mm\,day}^{-1}$) is simply the minimum of the available energy ($R_n$) and the current soil moisture state (SM in mm):

$$ET_{pot}^{<s,\,t>} = \min\left(R_n^{<s,\,t>}, SM^{<s,\,t-1>}\right) \tag{A7}$$

Soil evaporation ($E_s$ in $\mathrm{mm\,day}^{-1}$) is modeled as a function of vegetation, potential evapotranspiration (ET), and a NN learned parameter $\alpha_{Es}$ ($0 < \alpha_{Es} < 1$):

$$E_s^{<s,\,t>} = \left(1 - \mathrm{fPAR}^{<s,\,t>}\right) \cdot ET_{pot}^{<s,\,t>} \cdot \alpha_{E_s}^{<s,\,t>} \tag{A8}$$

Then, SM (mm) is updated as follows:

$$SM^{<s,\,t>} = SM^{<s,\,t-1>} - E_s^{<s,\,t>} \tag{A9}$$

Potential ET is updated again, using Eq. (A7). Transpiration $(\mathrm{mm\,day}^{-1})$ is represented in a similar way to soil evaporation (see Eq. (A8)):

$$T^{<s,\,t>} = \mathrm{fPAR}^{<s,\,t>} \cdot ET_{pot}^{<s,\,t>} \cdot \alpha_T^{<s,\,t>} \tag{A10}$$

where $0 < \alpha_T < 1$. SM is updated using transpiration:

$$SM^{<s,\,t>} = SM^{<s,\,t-1>} - T^{<s,\,t>} \tag{A11}$$

ET $(\mathrm{mm\,day}^{-1})$ is the sum of transpiration, soil and interception evaporation:

$$ET^{<s,\,t>} = E_i^{<s,\,t>} + E_s^{<s,\,t>} + T^{<s,\,t>} \tag{A12}$$

Note that, ET is constrained directly.

## A3 Soil and groundwater recharge

Water input ($w_{\text{in}}$ in $\mathrm{mm\,day^{-1}}$) is defined as the amount of water that arrives on the land surface:

$$w_{in}^{<s,\,t>} = \text{rainfall}^{<s,\,t>} + s_{melt}^{<s,\,t>} - E_i^{<s,\,t>} \tag{A13}$$

Soil recharge fraction (-) represents the fraction of incoming water that will be infiltrated to the soil:

$$r_{soil_{\text{fraction}}}^{<s,\,t>} = \min\left(1, \left(\frac{SM_{\max}^{<s>} - SM^{<s,\,t>}}{\max\left(w_{in}^{<s,\,t>},\,\epsilon\right)}\right)\right) \cdot \alpha_{r_{soil}}^{<s,\,t>} \tag{A14}$$

where, $SM_{\max}$ (mm) ($> 0$) is the maximum amount of water that can be held by the soil which is directly available to plants via transpiration and $\alpha_{\text{r}_{\text{soil}}}$ ($0 < \alpha_{r_{soil}} < 1$) represents uncertain processes. Both of these parameters are directly learned by NN. $\epsilon$ is a small value ($10^{-8}$) that is used to make the function differentiable under all circumstances, which is important for stable NN training. Incoming water and soil recharge fraction is used to model soil recharge ($\mathrm{mm\,day^{-1}}$):

$$r_{soil}^{<s,\,t>} = r_{soil_{\text{fraction}}}^{<s,\,t>} \cdot w_{in}^{<s,\,t>} \tag{A15}$$

Soil recharge infiltrates into the soil:

$$SM^{<s,\,t>} = SM^{<s,\,t>} + r_{soil}^{<s,\,t>} \tag{A16}$$

Groundwater recharge fraction (-) is modeled as a function of soil recharge fraction and a NN learned parameter $\alpha_{\text{r}_{\text{gw}}}$ ($0 < \alpha_{\text{r}_{\text{gw}}} < 1$):

$$r_{gw_{\text{fraction}}}^{<s,\,t>} = \left(1 - r_{soil_{\text{fraction}}}^{<s,\,t>}\right) \cdot \alpha_{r_{gw}}^{<s,\,t>} \tag{A17}$$

which is used to model groundwater recharge ($\mathrm{mm\,day^{-1}}$) defined as the amount of incoming water that will enter the groundwater:

$$r_{gw}^{<s,\,t>} = r_{gw_{\text{fraction}}}^{<s,\,t>} \cdot w_{in}^{<s,\,t>} \tag{A18}$$

## A4 Runoff

Soil recharge fraction and the NN learned parameter $\alpha_{\text{r}_{\text{gw}}}$ is used to model the fraction of surface runoff (-):

$$q_{surf_\text{fraction}}^{<s, t>} = \left(1 - r_{soil_\text{fraction}}^{<s, t>}\right) \cdot \left(1 - \alpha_{r_{gw}}^{<s, t>}\right) \tag{A19}$$

Surface runoff ($\mathrm{mm\,day^{-1}}$) refers to the amount of incoming water that becomes runoff:

$$q_{surf}^{<s, t>} = q_{surf_\text{fraction}}^{<s, t>} \cdot w_{in}^{<s, t>} \tag{A20}$$

Baseflow runoff ($\mathrm{mm\,day^{-1}}$) is defined as the total amount of water that is discharged from the groundwater:

$$q_{base}^{<s, t>} = GW^{<s, t-1>} \cdot \beta_{gw} \tag{A21}$$

where GW ($\mathrm{mm}$) is the current groundwater storage and $\beta_\text{gw}$ is a global constant that is directly learned by NN and refers to the baseflow recession.

Total runoff ($\mathrm{mm\,day^{-1}}$) is the sum of surface runoff and baseflow (and it is directly constrained):

$$q_{total}^{<s, t>} = q_{surf}^{<s, t>} + q_{base}^{<s, t>} \tag{A22}$$

### A5   Groundwater storage

Groundwater storage (GW in $\mathrm{mm}$) is updated as a function of the current GW, grundwater recharge, and baseflow as follows:

$$GW^{<s, t>} = GW^{<s, t-1>} + r_{gw}^{<s, t>} - q_{base}^{<s, t>} \tag{A23}$$

### A6   Terrestrial water storage

Terrestrial water storage (TWS in $\mathrm{mm}$) is the sum of all the modeled water storages:

$$TWS^{<s, t>} = SWE^{<s, t>} + GW^{<s, t>} + SM^{<s, t>} \tag{A24}$$

Note that, the modeled anomalies of TWS (not the raw simulations of TWS) is directly constrained.

### Appendix B: Model evaluation

### B1   Performance on SWE, ET, Runoff and fAPAR for major regions

This section shows the model performance with respect to the observation based SWE (Fig. B1) and fAPAR (for major regions) (Fig. B4), and ML based model constraints ET (Fig. B2), and Runoff (Fig. B3).

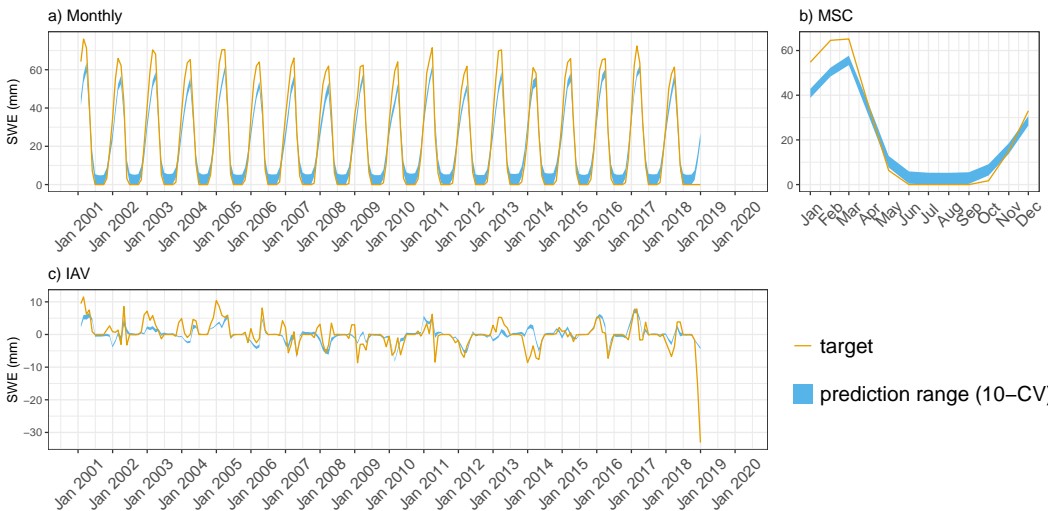

**Figure B1.** Predicted versus observed mean SWE over the testing set (spatial domain) across different folds: a) monthly, b) mean seasonal cycle and c) interannual variability

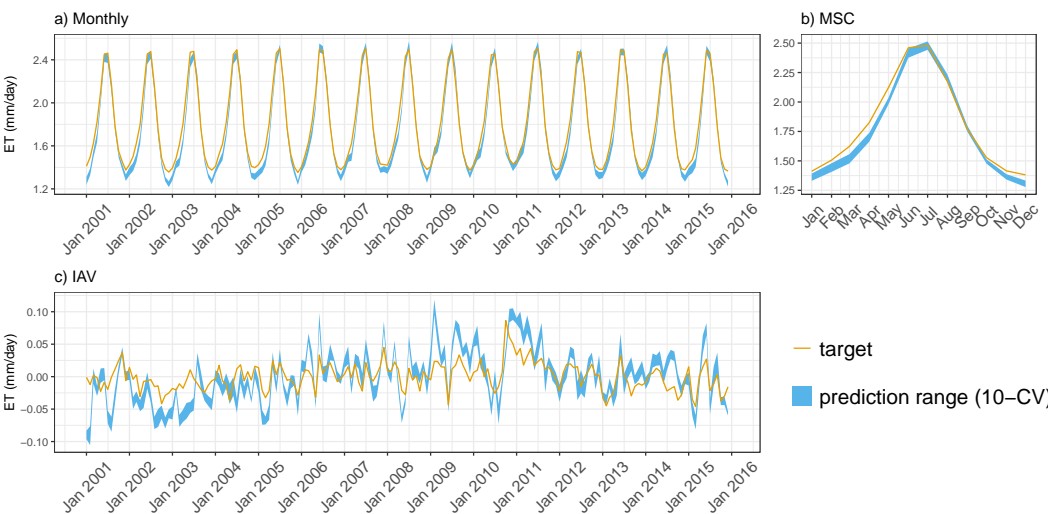

**Figure B2.** Predicted versus target mean ET over the testing set (spatial domain) across different folds: a) monthly, b) mean seasonal cycle and c) interannual variability

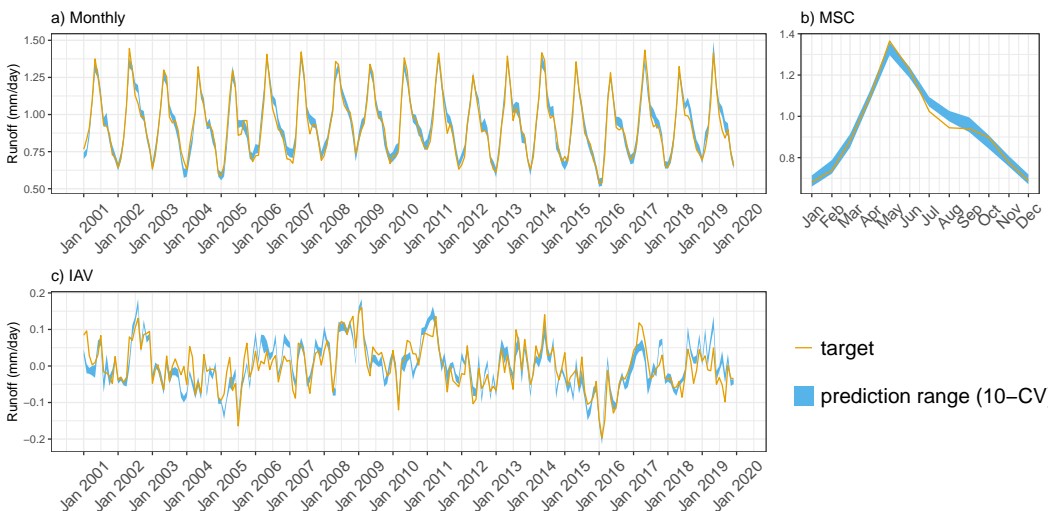

**Figure B3.** Predicted versus target mean Runoff over the testing set (spatial domain) across different folds: a) monthly, b) mean seasonal cycle and c) interannual variability.

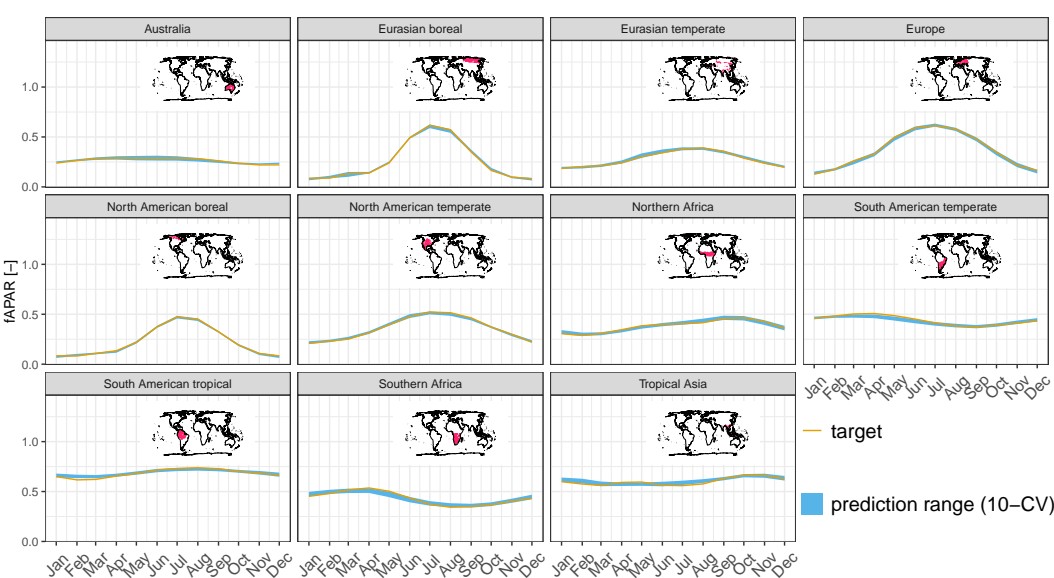

**Figure B4.** Predicted fAPAR (MSC) versus observations for major regions across different folds.

## B2 Global model performance

In this section, we demonstrate H2MV's performance on predicted global patterns and compare it to the performance of H2M (Kraft et al., 2022) (Fig. B5). Overall, our model performs slightly worse than H2M on the parameters TWS, SWE, and ET

in terms of RMSE, and on TWS and SWE in terms of SDR. We argue that this slight performance drop is expected because H2MV incorporates more physical formulations than H2M, resulting in stronger physical constraints, i.e., regularization. On the other hand, H2M has greater flexibility in adapting to the data and it likely explains the slight performance decline of H2MV compared to H2M.

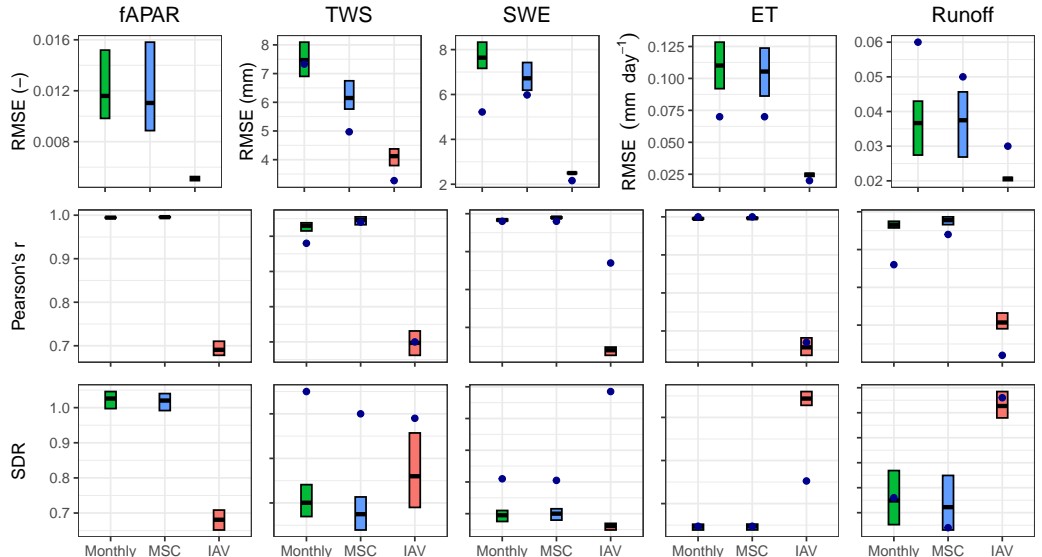

**Figure B5.** Model performance on the global data. Cross bars show the maximum and minimum error, and the lines show the mean error across 10 folds. The rows are metrics and the columns are model constraints. The dots show the model performance of H2M.

## Appendix C: Model generalizability in terms of space and time

In this section, we demonstrate the generalizability of our model across both spatial and temporal dimensions. We conducted an additional experiment by training the model while holding out the last five years of data to assess its spatio-temporal generalizability. Figure C1 illustrates the performance comparison between spatially split CV folds and spatio-temporal CV folds for time-series data post-2014. The "spatial split" refers to the model trained using the complete time-series data from 2001 to 2019, with only spatial grids held out. Conversely, the "spatio-temporal split" involves training the model with 14 years of time-series data (2001-2014) and holding out the data after 2014 for spatio-temporal testing. Overall, the results indicate that our model generalizes well both in time and space (Fig. C1). The performance is consistent across all parameters in both experiments, as measured by Pearson's r and SDR. The RMSE is also similar for all parameters, except for TWS. RMSE of TWS is slightly higher in the spatio-temporal split experiment (approximately 2 mm), while correlations remained consistent suggesting that the larger error is associated with larger variance of TWS. This suggests that the original model effectively generalizes across both temporal and spatial dimensions.

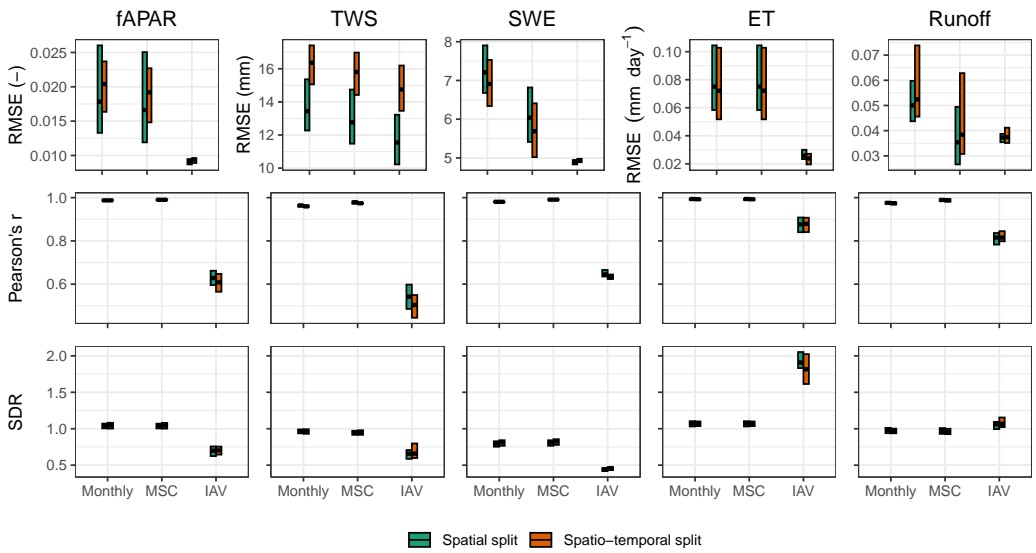

**Figure C1.** Model performance on the testing set based on the time-series data post-2014. 'Spatial split' refers to the model trained on the full time-series (2001-2019) while holding out spatial grids only. 'Spatio-temporal split' uses 14 years of data (2001-2014) for training, holding out post-2014 data for testing. Cross bars show the maximum and minimum error, and the lines show the mean error across 10 folds. The rows are metrics and the columns are model constraints. RMSE refers to root mean squared error, and SDR is standard deviation ratio (the ratio between predicted and observed standard deviation).

*Author contributions.* ZB implemented the model, performed the analysis and drafted the manuscript. MJ designed the water balance model
structure, and BK the initial deep learning architecture. All authors contributed intellectual input to the design, associated analysis, and writing.

*Competing interests.* The authors declare no competing interests.

*Acknowledgements.* We gratefully acknowledge financial support through the German Aerospace Center (DLR) with funds provided by the Federal Ministry for Economic Affairs and Climate Action (BMWK) due to an enactment of the German Bundestag under Grant
No. 50EE2209A. We further acknowledge the support by the European Research Council (ERC) Synergy Grant "Understanding and Modelling the Earth System with Machine Learning (USMILE)" under the Horizon 2020 research and innovation programme (Grant No. 855187). Zavud Baghirov is supported by the International Max Planck Research School for Global Biogeochemical Cycles (IMPRS-gBGC). We gratefully acknowledge the financial support from the Max Planck Society, which enabled us to publish this manuscript as open-access.

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
