# Peer review of "H2MV (v1.0): Global Physically-Constrained Deep Learning Water Cycle Model with Vegetation"

_EGUsphere, 2024_

## Referee Comment (RC2)

**Review of Manuscript**

**'H2MV (v1.0): Global Physically-Constrained Deep Learning Water Cycle Model with Vegetation'**

By Z. Baghirov et al.

Dear Editor,

I have reviewed the manuscript. My conclusions and comments are as follows:

**1. Scope**

The article is within the scope of GMD.

**2. Summary**

The authors present the H2MV model, a further development of the H2M model (Kraft et al., 2022). H2MV is a hybrid model for the global terrestrial water cycle. It consists of a conceptual process-based hydrological model including the main terrestrial stocks and fluxes of water, and connected neural networks (partly static, partly dynamic with memory) processing static catchment attributes and dynamic forcing to deliver space-variable, time-static catchment attributes (maximum vegetation-reachable soil water $SM_{max}$), space-time-variable vegetation states (fraction of absorbed photosynthetically active radiation fAPAR), and various space-time-variable parameters of the process-based hydrological model. Model forcing includes precipitation (P), radiation (Rad) and air temperature (T). Further observations used for model training are terrestrial water storage (TWS), fraction of absorbed photosynthetically active radiation (fAPAR), snow water equivalent (SWE) and runoff (Q). H2MV is trained in a Cross-validation (C-V) approach on 10 spatially mutually exclusive datasets, and validated on an additional spatial holdout set. Model performance is discussed for all predictive variables TWS, fAPAR, SWE and Q on various temporal aggregations (monthly, seasonal, interannual). The authors conclude that generally, model performance is acceptable and shows space-time patterns in agreement with expert expectations and the literature. Further, the authors discuss model equifinality, here expressed as the predictive variability of the target variables among the 10 C-V models. Here, the authors conclude that mainly soil-related parameters are uncertain, and that model errors are dominated by phase shifts.

**3. Evaluation**

Overall, the work presented by the authors is an interesting and relevant contribution to global land surface modeling. The presentation style is mainly clear and complete, and the conclusions are supported by the results. So there are only minor revisions required to increase clarity and completeness before publication.
"L"=line

L3 "… we explicitly represent vegetation states by the fraction of absorbed photosynthetically active radiation (fAPAR), and by the maximum soil moisture capacity ($SM_{max}$), …". This is misleading, as it suggests $SM_{max}$ represents a vegetation state. From the rest of the text, I take it that $SM_{max}$ is a spatially variable but temporally invariant representation of the maximum (vegetation-reachable) soil water content, i.e. a purely soil-related property. Therefore I suggest rephrasing with a better distinction between and explanation of the abiotic and biotic controls of soil water capacity.

L8 The authors use the term 'constrain' throughout the manuscript to refer to observables used in an objective function during model training/calibration. As not all readers will be familiar with this use of terms, I suggest adding a related clarification, e.g. in L44.

L37 "Hybrid (or differentiable) modeling aims to address this challenge". The sentence suggests that hybrid modeling is synonymous to differentiable modeling. This is not the case. There are hybrid modeling approaches that do not require differentiability of the process-based part, and not all differentiable model are hybrids. Therefore I suggest rephrasing.

L81 Please add a short information about the length of the available data.

L 165 C-V approach: The authors use 10 validation sets (and one common test data set common) that are mutually exclusive in terms of space, but not time. I understand that a time-exclusive C-V approach for the validation sets may not be possible due to limited data, but, as high spatial correlation may exist between validation and testing sets for the same time, the testing set should be differing from the validation sets in terms of both space and time. This will help to better assess the models space-time generalization capabilities. My suggestion: Train the model on all but the last two years. Use all but the last two years for space-only CV testing in the same way as done now. Use the last two years for space-time independent testing.

L 174 Loss function: Eq. (9) calculates the total loss over all observed targets (TWS, fAPAR, SWE, Q). The targets come with different units, so their influence on total L might be different. How is equal weighting of each target in L assured? Is the loss calculated from the Z-transformed data as in Kraft et al. (2022)?

L201 It is unclear to me what the authors mean by "each estimated process". Please add this information to the text. Also, in L208, the authors refer to "parameters" rather than "processes". Please clarify.

Figs 3, 4, C1, C2, C3. Please add a x-axis label to plots a)

L261 As H2MV is a further development of H2M, a performance comparison between the two is important. The authors provide this comparison in the Appendix in Fig. C5, but do not discuss it. In Fig. C5, it becomes apparent that H2MV performs worse than H2M in terms of at least
- RMSE for TWS, SWE, ET
- SDR for TWS, SWE

While I do not think that a new model generation needs to outperform a previous one for all metrics, the reader will benefit from a more detailed discussion of the performance differences between H2MV and H2M.

Parameter stability: I wonder how time-stable or time-variant the LSTM-predicted parameters of the hydrological model are. Ideally, if the hydrological model would fully contain all relevant processes, the parameters should be static. Time-variations would point at functional deficiencies of the hydrological model, and the time-patterns could point at the nature of these functional deficiencies. See e.g. Fig. 8 in Acuna Espinoza et al. (2024). I do not require that such a discussion is added to the current paper, rather it is a suggestion for further work.

Yours sincerely,

Uwe Ehret

**References**

Acuña Espinoza, E., Loritz, R., Álvarez Chaves, M., Bäuerle, N., and Ehret, U.: To bucket or not to bucket? Analyzing the performance and interpretability of hybrid hydrological models with dynamic parameterization, Hydrol. Earth Syst. Sci., 28, 2705–2719, https://doi.org/10.5194/hess-28-2705-2024, 2024.

Kraft, B., Jung, M., Körner, M., Koirala, S., and Reichstein, M.: Towards hybrid modeling of the global hydrological cycle, Hydrology and Earth System Sciences, 26, 1579–1614, 2022.

---

## Author Response (AR1)

**Author's response to reviewers**

We appreciate the detailed reviews provided by both reviewers, which have significantly contributed to improving the clarity of our manuscript.

We display the reviewers' comments in black and italic and highlight our responses in green in order to ensure clarity.

Best regards, Zavud Baghirov on behalf of the authors

**Reviewer 1**

**General evaluation:**

*This paper presents "H2MV", a Global Physically-Constrained Deep Learning Water Cycle Model with Vegetation. The proposed hybrid hydrological model extends a traditional physics-based hydrological model, combined with a static and a dynamic data-driven module. These modules learn temporal parameters (coefficients) as well as spatially-varying parameters. The novelty lies in explicitly representing vegetation constraints that play a relevant role in the water cycle, with the goal to achieve a more realistic and interpretable partitioning of evapotranspiration. The model is optimized against several observation products, and is tested for equifinality. The model code is publicly available.*

*Overall, the paper presents a valuable advance in hydrological modelling and is generally well written. However, the placement into existing literature could be more comprehensive. Further, the discussion of results is somewhat hard to follow, since the "analysis route" is not laid out beforehand, and many abbreviations destroy the flow.*

*With some minor edits, I believe this manuscript would qualify for publication with GMD. Please find my comments below.*

We acknowledge that our manuscript required improvements, particularly in terms of clarity. We have made improvements to the literature review, methods section, and results. Below are the point-by-point changes we implemented with respect to the reviewer 1.

**Specific comments:**

*l. 37: "Hybrid (or differentiable) modeling..." – I would see differentiable modeling as a specific case of hybrid modeling. Please be more specific.*

We agree with the reviewer that this sentence was misleading. We removed "(or differentiable)" to prevent any potential misunderstanding.

*l. 40 ff.: There are many more studies that demonstrate the use of LSTMs (or other networks) to "hybridize" physics-based hydrological models. Please provide a brief but broader overview of the literature before going into the specific model versions by Kraft et al. that you aim to improve here. (Part of Section 3.4 addresses this to some extent – consider moving this discussion of existing approaches to the introduction.)*

We relocated the discussion of the existing literature from Section 3.4 to the introduction. Additionally, we expanded it by including recent studies on the topic. Below is the discussion of the existing approaches:

"Recent studies have been exploring the integration of process knowledge into machine learning models to better constrain uncertain processes with hybrid approaches. For instance, Zhao et al. (2019) developed a hybrid model that merges a NN with an evapotranspiration model to estimate latent heat flux, ensuring it adheres to the conservation of energy principle. This model performed better in extrapolating beyond the data range of the training set, compared to a more data-driven model. Similarly, ElGhawi et al. (2023) combined NN with a mechanistic latent heat flux model to estimate the surface and aerodynamic resistances of vegetation. While their model successfully estimated latent heat flux, it faced the challenge of equifinality. To address this, they applied both theoretical and data constraints. In a comparable effort, Koppa et al. (2022) utilized a process-based model of terrestrial evaporation alongside NN to estimate transpiration stress. Zhong, Lei, and Gao (2023) integrated deep learning with a hydrological model to estimate runoff changes, demonstrating that this approach enhances the reliability of projections in permafrost-affected mountain headwaters. Similarly, Bennett and Nijssen (2021) combined neural networks with a process-based hydrological model to simulate turbulent heat fluxes, concluding that this method offers advantages over both purely process-based models and purely machine learning-based estimations. Additionally, Bhasme, Vagadiya, and Bhatia (2022) merged neural networks with a conceptual hydrological model to effectively estimate evapotranspiration and streamflow in regional catchments."

*Section 2.1: The introduction to the dataset is very short. Please mention at least which inputs are used, and provide a little context.*

We extended the discussion of the dataset used in Section 2.1. We provided a brief explanation of the inputs and observational data, and included the time periods of the datasets we used. Below is the improved version of the Section 2.1:

"We use meteorological time series data—specifically precipitation, net radiation, and air temperature—as temporal inputs (forcing) for our model. In addition to these temporal inputs, we incorporate static features such as soil properties, land cover fractions, elevation, and wetlands as static inputs.

Our model is optimized against observations of terrestrial water storage, fAPAR, and snow water equivalent, as well as observation-based estimations of evapotranspiration and runoff. We refer to these observations as "constraints", as they guide or limit the model's behavior.

Table 1 shows the detailed information about the used datasets. All meteorological forcing and model constraints were aggregated to 1° spatial resolution. The spatial resolutions of static inputs were aggregated to 1/30°. We use compressed representations of the original static input that was preprocessed in a separate modeling framework (for details Kraft et al. (2022) can be referred). Meteorological forcing and SWE are kept in the native daily temporal resolutions, while monthly temporal resolution is used for the rest of the model constraints.

Temporal coverage of the data we use vary:

- Meteorological forcing (all): 2001 to 2019 (19 years of daily data)
- TWS: 2001 to 2017 (17 years of monthly data)
- fAPAR: 2001 to 2019 (19 years of monthly data)
- SWE: 2001 to 2018 (18 years of daily data)
- ET: 2001 to 2015 (15 years of monthly data)
- Runoff: 2001 to 2019 (19 years of monthly data)"

*Eq. 2: Since fAPAR and all alphas are space- and time-dependent and are multiplied with each other, could you comment on the expected identifiability of your introduced parameters?*

We acknowledge the importance of addressing identifiability (or equifinality) in our introduced parameters. Equifinality presents a challenge in hybrid modeling, as it does in other methodologies, including purely process-based approach. Therefore, we explore this aspect in the manuscript and estimate equifinality using a straightforward cross-validation approach. While we do not directly focus on the alphas in the manuscript, we do analyze the parameters depending on the alphas and discuss their identifiability in detail in Section 3.2.

Regarding fAPAR predictions, we do not expect equifinality, as the model directly constrains these predictions using fAPAR observations. In the revised manuscript, we have extended the paragraph around line 350 with additional notes on the identifiability of the parameters used in Equations 2, 3, and 4.

"... Notably, transpiration is predicted more robustly compared to interception and soil evaporation, indicating that equifinality of ET partitioning is primarily between soil and interception evaporation. This indicates that the identifiability of the neural network's learned parameters for evapotranspiration partitioning is weak, particularly for the parameters used to model soil and interception evaporation."

*Given the different alphas and their individual constraints, is there still a mass balance constraint over the entire hydrological model, or is this given up in the spirit of "local adjustments"?*

This is an important point and we agree that it needs to be explicitly discussed in the manuscript. When designing the conceptual hydrological model, we ensured it fully adheres

to the mass balance principle. The proposed hydrological model equations guarantee that the amount of water entering the system (i.e., precipitation) equals the amount of water leaving the system plus any change in storage within the system. This constraint ensures that the neural networks also adhere to the mass balance principle. This is why we find that our predictions cannot be perfectly fitted to the observational data (e.g., terrestrial water storage), despite neural networks being highly data-adaptive. We added the following paragraph in the Section 2.2.1 (line 166):

"The proposed hydrological model equations ensure that the amount of water entering the system (i.e., precipitation) equals the amount of water leaving the system (i.e., evapotranspiration and runoff) plus any change in storage within the system. This constraint ensures that the neural networks also adhere to the principle of mass balance (Appendix A)."

*Section 2.4, "Model Evaluation": It should be mentioned here that for model performance evaluation, RMSE, Pearson's r and SDR are used.*

We added a new subsection (2.4.1) to the Section 2.4 titled as "Performance metrics", with the following short paragraph:

"We evaluate our model's performance using root mean square error (RMSE), Pearson's correlation coefficient (r), and the standard deviation ratio (SDR), which is the ratio between the predicted and observed standard deviations."

*Section 2.4.2: It is not clear why you are particularly interested in TWS, and a decomposition thereof. Please add a motivation here to guide the reader in your model evaluation efforts. In fact, after reading through the full manuscript, it seems this decomposition is never used. Omit this completely or include related results. Overall, the evaluation strategy should be much better explained here, see related comments below.*

Thank you for the feedback on the model evaluation section of the manuscript. TWS anomaly observations from GRACE are one of our main constraints because these are direct satellite observations of an integrated hydrological state available globally and time-resolved. In our model, TWS is the sum of three main water storages that lack direct observations (except for snow water equivalent, which is derived from observational data). Therefore, it is important and interesting to assess which component of TWS is more dominant and where (spatially), in particular because previous studies highlighted large modelling uncertainties associated to this. We added the following paragraph to the manuscript (Section 2.4.5):

"In our model, TWS is composed of three primary water storage components that are not directly observed, except for snow water equivalent, which is derived from observational data. It is crucial and intriguing to evaluate which component of TWS is most dominant and where this dominance occurs spatially. This is particularly important because previous studies have highlighted significant modeling uncertainties related to these components (Trautmann et al. 2018; Kraft et al. 2022)."

Regarding the placement of results and discussion: Section 2.4.2 introduces the method for decomposing TWS, and we discuss the corresponding results in Section 3.2.5. Section 2.4.2 is

titled "TWS decomposition" while Section 3.2.5 is titled "Water storage decomposition". This discrepancy on the titles may be causing confusion, and we changed the title of Section 3.2.5 (3.3.5 in the revised version) to "TWS decomposition" for clarity.

*Also, in the results part, IAV plays a dominant role, but has never been introduced.*

We added 2 subsections "Mean seasonal cycle" (2.4.2) and "Interannual variability" (2.4.3) where we define the terms:

"We define the mean seasonal cycle (MSC) as follows:

$$MSC\,(m) \;=\; \frac{1}{Y}\,\sum_{y=1}^{Y} p_{m,y} \tag{1}$$

where $p_{m,y}$ represents the modeled or observed parameter for month $m$ and year $y$, and $Y$ is the total number of years.

We define the interannual variability (IAV) as follows:

$$IAV\,(m,\;y) \;=\; p_{m,y} \;-\; MSC\,(m) \tag{2}$$

In this equation, $p_{m,y}$ denotes the modeled or observed parameter for a given month $m$ and year $y$, while $MSC(m)$ represents the mean seasonal cycle for that specific month $m$."

*It would make sense to bring Fig. B1 to the main body of the manuscript.*

Agreed. We moved the Fig. B1 to the main body of the manuscript.

*Fig. 5 is referenced before Fig. 3, so reorder the Figures, or reorder the storyline of the results Section (Fig. 5 is easier to understand after the discussion of Figs. 3 and 4).*

We reordered the storyline of the results section. We reference and discuss Figures 3 and 4 before discussing Figure 5 now. (Please note that these are Figures 4, 5 and 6 in the new version of the manuscript).

*Fig. 3: How is the target obtained? Is it the mean value over all spatial domains contained in the testing set? Why is the variability of the target not shown? That would help judge the credibility of the model. And/or show individual spatial regions.*

Indeed, the target in Fig. 3 (Fig. 4 in the new version) represents the observed mean fAPAR value over the testing set (spatial domain). We added this information to the respective figure captions:

[Figure]

Figure 1: Predicted versus observed mean fAPAR over the testing set (spatial domain) across different folds: a) monthly, b) mean seasonal cycle and c) interannual variability.

The main purpose of this figure is to evaluate the general performance of our model on the testing set. We actually display our model's performance for fAPAR across the major regions in Fig. B4.

*Fig. 8: The term "Equifinality index" is used here for the first time. Introduce in Section 2.4.1 or replace the term in the results part.*

We defined the term "equifinality index" in Section 2.4.4:

"We define equifinality index (EI) as follows:

$$EI = e_{\text{phase}} + e_{\text{bias}} + e_{\text{var}} \tag{3}$$

EI is essentially MSE normalised by variance of the estimations. Higher EI values signify a larger degree of equifinality, or reduced robustness, while lower values indicate smaller equifinality, and therefore a more robust prediction."

*I would have expected a discussion of equifinality (Section 3.3) before Section 3.2 (interpretation of results with respect to emerging global patterns), and a judgement of these interpretations based on the findings about equifinality (how robust are the conclusions you draw in Section 3.2, given that some states are not perfectly constrained?).*

We moved the discussion of equifinality (now Section 3.2) before "Emerging global patterns" (now Section 3.3).

**Technical comments:**

- *l. 32: maybe "physics-based" instead of "physical" (they are still computer models…)*

- *l. 41: "a ... network" instead of "a ... networks"*
- *l. 230 "interannual" instead of "interranual" (several instances throughout the manuscript, and other misspellings such as "interannaul")*
- *l. 238: "observed patterns ... are" instead of "is"*

We corrected all of these points.

**Reviewer 2 (Uwe Ehret)**

**1. Scope**

*The article is within the scope of GMD.*

**2. Summary**

*The authors present the H2MV model, a further development of the H2M model (Kraft et al., 2022). H2MV is a hybrid model for the global terrestrial water cycle. It consists of a conceptual process based hydrological model including the main terrestrial stocks and fluxes of water, and connected neural networks (partly static, partly dynamic with memory) processing static catchment attributes and dynamic forcing to deliver space-variable, time-static catchment attributes (maximum vegetation-reachable soil water SMmax), space-time-variable vegetation states (fraction of absorbed photosynthetically active radiation fAPAR), and various space-time-variable parameters of the process-based hydrological model. Model forcing includes precipitation (P), radiation (Rad) and air temperature (T). Further observations used for model training are terrestrial water storage (TWS), fraction of absorbed photosynthetically active radiation (fAPAR), snow water equivalent (SWE) and runoff (Q). H2MV is trained in a Cross-validation (C-V) approach on 10 spatially mutually exclusive datasets, and validated on an additional spatial holdout set. Model performance is discussed for all predictive variables TWS, fAPAR, SWE and Q on various temporal aggregations (monthly, seasonal, interannual). The authors conclude that generally, model performance is acceptable and shows space-time patterns in agreement with expert expectations and the literature. Further, the authors discuss model equifinality, here expressed as the predictive variability of the target variables among the 10 C-V models. Here, the authors conclude that mainly soil-related parameters are uncertain, and that model errors are dominated by phase shifts.*

**3. Evaluation**

*Overall, the work presented by the authors is an interesting and relevant contribution to global land surface modeling. The presentation style is mainly clear and complete, and the conclusions are supported by the results. So there are only minor revisions required to increase clarity and completeness before publication.*

Below are the point-by-point changes we implemented with respect to the reviewer 2 (Uwe Ehret).

*L3 "... we explicitly represent vegetation states by the fraction of absorbed photosynthetically active radiation (fAPAR), and by the maximum soil moisture capacity (SMmax), ...". This is*

*misleading, as it suggests SMmax represents a vegetation state. From the rest of the text, I take it that SMmax is a spatially variable but temporally invariant representation of the maximum (vegetation-reachable) soil water content, i.e. a purely soil-related property. Therefore I suggest rephrasing with a better distinction between and explanation of the abiotic and biotic controls of soil water capacity.*

We agree that the sentence could be misleading. Indeed, we define $SM_{max}$ as spatially varying and temporally constant. It represents the maximum amount of water that can be stored in the soil and transpired by vegetation. Therefore, while $SM_{max}$ is closely related to vegetation in our model, it does not only represent a vegetation state (effective rooting depth) but also depends on soil properties. We clarified this in the revised manuscript by rephrasing the sentence (L3):

In the hydrological model, vegetation states are represented by the fraction of absorbed photosynthetically active radiation (fAPAR), and soil storage capacity ($SM_{max}$), which depends on effective rooting depth besides soil properties. $SM_{max}$ and fAPAR are both learned and predicted by the neural networks directly.

*L8 The authors use the term 'constrain' throughout the manuscript to refer to observables used in an objective function during model training/calibration. As not all readers will be familiar with this use of terms, I suggest adding a related clarification, e.g. in L44.*

We added the following sentence in line 58:

"... The study employed observational products of TWS variations, snow, ET, and runoff to constrain (i.e., to calibrate) the model."

Additionally, we included the following in line 101:

"... We refer to these observations as"constraints", as they confine the model's behavior to the observed patterns."

*L37 "Hybrid (or differentiable) modeling aims to address this challenge". The sentence suggests that hybrid modeling is synonymous to differentiable modeling. This is not the case. There are hybrid modeling approaches that do not require differentiability of the process-based part, and not all differentiable model are hybrids. Therefore I suggest rephrasing.*

Agreed and we removed "(or differentiable)" to prevent any potential misunderstanding in the revised version of the manuscript.

*L81 Please add a short information about the length of the available data.*

We added the following to the Section 2.1 to explain the periods of the data:

"Temporal coverage of the data we use vary:

- Meteorological forcing (all): 2001 to 2019 (19 years of daily data)
- TWS: 2001 to 2017 (17 years of monthly data)
- fAPAR: 2001 to 2019 (19 years of monthly data)

- SWE: 2001 to 2018 (18 years of daily data)
- ET: 2001 to 2015 (15 years of monthly data)
- Runoff: 2001 to 2019 (19 years of monthly data)"]

*L165 C-V approach: The authors use 10 validation sets (and one common test data set common) that are mutually exclusive in terms of space, but not time. I understand that a time-exclusive C-V approach for the validation sets may not be possible due to limited data, but, as high spatial correlation may exist between validation and testing sets for the same time, the testing set should be differing from the validation sets in terms of both space and time. This will help to better assess the models space-time generalization capabilities. My suggestion: Train the model on all but the last two years. Use all but the last two years for space-only CV testing in the same way as done now. Use the last two years for space-time independent testing.*

We have conducted the experiment as suggested by the reviewer. Specifically, we used the data between 2001 and 2014 for training, and held out the data post 2014 for spatio-temporal testing. The results are qualitatively consistent with our original spatial cross-validation strategy and we added the discussion of this result in the Appendix C:

"In this section, we demonstrate the generalizability of our model across both spatial and temporal dimensions. We conducted an additional experiment by training the model while holding out the last five years of data to assess its spatio-temporal generalizability.

Figure 2 (Fig. C1 in the revised manuscript) illustrates the performance comparison between spatially split CV folds and spatio-temporal CV folds for time-series data post-2014. The "spatial split" refers to the model trained using the complete time-series data from 2001 to 2019, with only spatial grids held out. Conversely, the "spatio-temporal split" involves training the model with 14 years of time-series data (2001-2014) and holding out the data after 2014 for spatio-temporal testing.

Overall, the results indicate that our model generalizes well both in time and space. The performance is consistent across all parameters in both experiments, as measured by Pearson's r and SDR. The RMSE is also similar for all parameters, except for TWS. RMSE of TWS is slightly higher in the spatio-temporal split experiment (approximately 2 mm), while correlations remained consistent suggesting that the larger error is associated with larger variance of TWS. This suggests that the original model effectively generalizes across both temporal and spatial dimensions."

[Figure]

Figure 2: Model performance on the testing set based on the time-series data post-2014. 'Spatial split' refers to the model trained on the full time-series (2001-2019) while holding out spatial grids only. 'Spatio-temporal split' uses 14 years of data (2001-2014) for training, holding out post-2014 data for testing. Cross bars show the maximum and minimum error, and the lines show the mean error across 10 folds. The rows are metrics and the columns are model constraints. RMSE refers to root mean squared error, and SDR is standard deviation ratio (the ratio between predicted and observed standard deviation).

*L 174 Loss function: Eq. (9) calculates the total loss over all observed targets (TWS, fAPAR, SWE, Q). The targets come with different units, so their influence on total L might be different. How is equal weighting of each target in L assured? Is the loss calculated from the Z-transformed data as in Kraft et al. (2022)?*

Indeed, different targets have different units, so we standardize them using Z transformation before calculating each individual loss and then summing all the loss terms. We clarified this in the revised version of the manuscript, by adding the following to the Section 2.3.1:

"We apply a Z-transformation to both the predictions and their corresponding observations before calculating the loss. This addresses the issue of differing units among model constraints and ensures that each individual loss has a similar impact on the model's behavior."

*L201 It is unclear to me what the authors mean by "each estimated process". Please add this information to the text. Also, in L208, the authors refer to "parameters" rather than "processes". Please clarify.*

We agree that it is confusing. In the revised version of the manuscript, we consistently use the term "estimated parameters" for clarity.

*Figs 3, 4, C1, C2, C3. Please add a x-axis label to plots a)*

Done.

*L261 As H2MV is a further development of H2M, a performance comparison between the two is important. The authors provide this comparison in the Appendix in Fig. C5, but do not discuss it. In Fig. C5, it becomes apparent that H2MV performs worse than H2M in terms of at least - RMSE for TWS, SWE, ET - SDR for TWS, SWE While I do not think that a new model generation needs to outperform a previous one for all metrics, the reader will benefit from a more detailed discussion of the performance differences between H2MV and H2M.*

Indeed, H2MV performs slightly worse than H2M on these parameters. We added the following paragraph in Appendix B2 to explain why we think this is plausible:

"In this section, we demonstrate H2MV's performance on predicted global patterns and compare it to the performance of H2M (Kraft et al. 2022). Overall, our model performs slightly worse than H2M on the parameters TWS, SWE, and ET in terms of RMSE, and on TWS and SWE in terms of SDR. We argue that this slight performance drop is expected because H2MV incorporates more physical formulations than H2M, resulting in stronger physical constraints, i.e., regularization. On the other hand, H2M has greater flexibility in adapting to the data and it likely explains the slight performance decline of H2MV compared to H2M."

*Parameter stability: I wonder how time-stable or time-variant the LSTM-predicted parameters of the hydrological model are. Ideally, if the hydrological model would fully contain all relevant processes, the parameters should be static. Time-variations would point at functional deficiencies of the hydrological model, and the time-patterns could point at the nature of these functional deficiencies. See e.g. Fig. 8 in Acuna Espinoza et al. (2024). I do not require that such a discussion is added to the current paper, rather it is a suggestion for further work.*

We agree on this general consideration and that it deserves further attention. We are planning to address this in our future work.

**References**

Bennett, Andrew, and Bart Nijssen. 2021. "Deep Learned Process Parameterizations Provide Better Representations of Turbulent Heat Fluxes in Hydrologic Models." *Water Resources Research* 57 (5): e2020WR029328.

Bhasme, Pravin, Jenil Vagadiya, and Udit Bhatia. 2022. "Enhancing Predictive Skills in Physically-Consistent Way: Physics Informed Machine Learning for Hydrological Processes." *Journal of Hydrology* 615: 128618.

ElGhawi, RedaReda, Basil Kraft, Christian Reimers, Markus Reichstein, Marco Körner, Pierre Gentine, and Alexander J WinklerWinkler. 2023. "Hybrid Modeling of Evapotranspiration: Inferring Stomatal and Aerodynamic Resistances Using Combined Physics-Based and Machine Learning." *Environmental Research Letters* 18 (3): 034039.

Koppa, Akash, Dominik Rains, Petra Hulsman, Rafael Poyatos, and Diego G Miralles. 2022. "A Deep Learning-Based Hybrid Model of Global Terrestrial Evaporation." *Nature Communications* 13 (1): 1912.

Kraft, Basil, Martin Jung, Marco Körner, Sujan Koirala, and Markus Reichstein. 2022. "Towards Hybrid Modeling of the Global Hydrological Cycle." *Hydrology and Earth System Sciences* 26 (6): 1579–1614.

Trautmann, Tina, Sujan Koirala, Nuno Carvalhais, Annette Eicker, Manfred Fink, Christoph Niemann, and Martin Jung. 2018. "Understanding Terrestrial Water Storage Variations in Northern Latitudes Across Scales." *Hydrology and Earth System Sciences* 22 (7): 4061–82.

Zhao, Wen Li, Pierre Gentine, Markus Reichstein, Yao Zhang, Sha Zhou, Yeqiang Wen, Changjie Lin, Xi Li, and Guo Yu Qiu. 2019. "Physics-Constrained Machine Learning of Evapotranspiration." *Geophysical Research Letters* 46 (24): 14496–507.

Zhong, Liangjin, Huimin Lei, and Bing Gao. 2023. "Developing a Physics-Informed Deep Learning Model to Simulate Runoff Response to Climate Change in Alpine Catchments." *Water Resources Research* 59 (6): e2022WR034118.